# Magnetic field reveals vanishing Hall response in the normal state of stripe-ordered cuprates

Zhenzhong Shi [1,2], P. G. Baity [1,3,5], J. Terzic [1], Bal K. Pokharel[1,3], T. Sasagawa [4] & Dragana Popović [1,3 ✉]

The origin of the weak insulating behavior of the resistivity, i.e. $\rho_{xx} \propto \ln(1/T)$, revealed when magnetic fields ($H$) suppress superconductivity in underdoped cuprates has been a longtime mystery. Surprisingly, the high-field behavior of the resistivity observed recently in charge- and spin-stripe-ordered La-214 cuprates suggests a metallic, as opposed to insulating, high-field normal state. Here we report the vanishing of the Hall coefficient in this field-revealed normal state for all $T < (2-6)T_c^0$, where $T_c^0$ is the zero-field superconducting transition temperature. Our measurements demonstrate that this is a robust fundamental property of the normal state of cuprates with intertwined orders, exhibited in the previously unexplored regime of $T$ and $H$. The behavior of the high-field Hall coefficient is fundamentally different from that in other cuprates such as $YBa_2Cu_3O_{6+x}$ and $YBa_2Cu_4O_8$, and may imply an approximate particle-hole symmetry that is unique to stripe-ordered cuprates. Our results highlight the important role of the competing orders in determining the normal state of cuprates.

[1] National High Magnetic Field Laboratory, Florida State University, Tallahassee, FL 32310, USA. [2] School of Physical Science and Technology & Institute for Advanced Study, Soochow University, Suzhou 215006, China. [3] Department of Physics, Florida State University, Tallahassee, FL 32306, USA. [4] Materials and Structures Laboratory, Tokyo Institute of Technology, Kanagawa 226-8503, Japan. [5]Present address: James Watt School of Engineering, University of Glasgow, Glasgow G12 8QQ, Scotland, United Kingdom. ✉email: dragana@magnet.fsu.edu

The central issue for understanding the high-temperature superconductivity in cuprates is the nature of the ground state that would have appeared had superconductivity not intervened. Therefore, magnetic fields have been commonly used to suppress superconductivity and expose the properties of the normal state, but the nature of the high-$H$ normal state may be further complicated by the interplay of charge and spin orders with superconductivity. La$_{2-x-y}$Sr$_x$(Nd,Eu)$_y$CuO$_4$ compounds are ideal candidates for probing the nature of the field-revealed ground state[1] of underdoped cuprates in the presence of inter-twined orders because, for doping levels near $x = 1/8$, they exhibit both spin and charge orders with the strongest correlations and lowest $T_c^0$ already at $H = 0$. In particular, in each CuO$_2$ plane, charge order appears in the form of static stripes that are separated by charge-poor regions of oppositely phased antiferromagnetism[2], i.e. spin stripes, with the onset temperatures $T_{CO} > T_{SO} > T_c^0$; stripes are rotated by 90° from one layer to next. The low values of $T_c^0$ have made it possible to determine the in-plane $T$–$H$ vortex phase diagram[3] using both linear and nonlinear transport over the relatively largest range of $T$ and perpendicular $H$ (i.e., $H\perp$ CuO$_2$ layers), and to probe deep into the high-field normal state. The most intriguing question, indeed, is what happens after the superconductivity is suppressed by $H$, i.e. for fields greater than the quantum melting field of the vortex solid where $T_c(H) \rightarrow 0$. It turns out that a wide regime of vortex liquid-like behavior, i.e. strong superconducting (SC) phase fluctuations, persists in two-dimensional (2D) CuO$_2$ layers, all the way up to the upper critical field $H_{c2}$. It is in this regime that recent electrical transport measurements have also revealed[4] the signatures of a spatially modulated SC state referred to as a pair density wave[5] (PDW). The normal state, found at $H > H_{c2}$, is highly anomalous[3]: it is characterized by a weak, insulating $T$-dependence of the in-plane longitudinal resistivity, $\rho_{xx} \propto \ln(1/T)$, without any sign of saturation down to at least $T/T_c^0 \sim 10^{-2}$, and the negative magnetoresistance (MR). In contrast to the $H$-independent $\ln(1/T)$ reported[6,7] for the case where there is no clear evidence of charge order[8] in $H = 0$ and where the high-$H$ normal state appears to be an insulator[6,9] here the $\ln(1/T)$ behavior is suppressed by $H$, strongly suggesting that $\rho_{xx}$ becomes independent of $T$, i.e., metallic, at high enough magnetic field ($H > 70$ T). In either case, the origin of such a weak, insulating behavior is not understood[7,10–13], but it is clear that the presence of stripes seems to affect the nature of the normal state. Therefore, additional experiments are needed to probe the highest-$H$ regime.

In cuprates, the Hall effect has been a powerful probe of the $T = 0$ field-revealed normal state (e.g. refs. [14–19] and refs. therein). In the high-field limit as $T \rightarrow 0$, the Hall coefficient $R_H$, obtained from the Hall resistivity $\rho_{yx}(H) = R_H H$, can be used to determine the sign and the density ($n$) of charge carriers. In a single-band metal, for example, $n = n_H$, where the Hall number $n_H = 1/(eR_H)$ and $e$ is the electron charge ($R_H > 0$ for holes, $R_H < 0$ for electrons). In general, the magnitude of $R_H$ reflects the degree of particle-hole asymmetry and, thus, understanding the Hall coefficient provides deep insight into the microscopic properties. However, the interpretation of the Hall effect in cuprates has been a challenge, because $R_H$ can depend on both $T$ and $H$, and it can be affected by various factors, such as the presence of SC correlations and the topological structure of the Fermi surface. For example, a drop of $R_H$ from positive to negative values with decreasing $T$, observed in underdoped cuprates for dopings where charge orders are present[20] at high $H$, has been attributed[14,19,21,22] to the Fermi surface reconstruction, which includes the appearance of electron pockets in the Fermi surface of a hole-doped cuprate. A drop in the normal state, positive $R_H(T)$, is, in fact, observed in all hole-doped cuprates near $x = 1/8$ (see ref. [21] and refs. therein). Other studies of the Hall effect in cuprates have focused on the

effects of SC fluctuations (refs. [23,24] and refs. therein), and on the pronounced change in the Hall number across the charge order and the pseudogap quantum critical points[16–18,25,26]. However, the Hall behavior in the $T \rightarrow 0$, $H > H_{c2}$ regime has remained mostly unexplored. In particular, recent studies of the La$_{2-x-y}$Sr$_x$(Nd,Eu)$_y$CuO$_4$ compounds have demonstrated[3,4] that reliable extrapolations to the $T \rightarrow 0$ normal state can be made only by tracking the evolution of SC correlations down to $T \ll T_c^0$ and $H/T_c^0$ [T/K] $\gg 1$, but there have been no studies of the Hall effect in stripe-ordered cuprates that extend to that regime of $T$ and $H$ and, specifically, to the anomalous normal state at $H > H_{c2}$.

Therefore, we measure the Hall effect on La$_{1.7}$Eu$_{0.2}$Sr$_{0.1}$CuO$_4$ and La$_{1.48}$Nd$_{0.4}$Sr$_{0.12}$CuO$_4$ (see Methods) over the entire in-plane $T$-$H$ vortex phase diagram previously established[3,4] for $T$ down to $T/T_c^0 \lesssim 0.003$ and fields up to $H/T_c^0 \sim 10$ T/K, and deep into the normal state. Combining the results of several techniques allows us to achieve an unambiguous interpretation of the Hall data for $H < H_{c2}$, and reveal novel properties of the normal state for $H > H_{c2}$. Our main results are summarized in the $T$-$H$ phase diagrams shown in Fig. 1. The key finding is that, in the high-field limit, the positive $R_H$ decreases to zero at $T = T_0(H)$ upon cooling, and it remains zero (see Methods) all the way down to the lowest measured $T$, despite the absence of any observable signs of superconductivity. Here, $T_0(H) \sim (2 - 3)T_c^0$ for La$_{1.7}$Eu$_{0.2}$Sr$_{0.1}$CuO$_4$ and $T_0(H) \sim 6T_c^0$ for La$_{1.48}$Nd$_{0.4}$Sr$_{0.12}$CuO$_4$. $T_c^0$, where the linear resistivity $\rho_{xx}$ becomes zero, and other characteristic temperatures, such as the pseudogap $T_{PG}$, are summarized in Table 1. Therefore, the vanishing Hall coefficient appears well below $T_{PG}$, in the temperature region where both charge and spin orders (i.e., stripes) have fully developed. Meanwhile, we note that the drop of $R_H$ at $T > T_0$ does not depend on $H$, while $T_0(H)$ is very weakly dependent on $H$ (Fig. 1), almost constant, suggesting that $R_H \approx 0$ is characteristic of the zero-field (normal) ground state in the presence of stripes.

## Results

**Hall coefficient**. Our main results are shown in Fig. 1. From the Hall measurements, we are able to identify regions in $(T, H)$ phase space with different signs of $R_H$ (Fig. 1a, b) and, in particular, we find $R_H \approx 0$ over a wide range of $T$ and $H$ in both materials. Further insight is obtained by comparing the Hall results with the phase diagram obtained by other transport techniques, as shown in Fig. 1c, d. The measurements of the in-plane magnetoresistance $\rho_{xx}(H)$ at different $T$ were used to determine[3,4] $T_c(H)$, the melting temperature of the vortex solid in which $\rho_{xx} = 0$. Although the quantum melting fields of the vortex solid are relatively low (~5.5 T and ~4 T, respectively, for La$_{1.7}$Eu$_{0.2}$Sr$_{0.1}$CuO$_4$ and La$_{1.48}$Nd$_{0.4}$Sr$_{0.12}$CuO$_4$), the regime of strong 2D SC phase fluctuations (vortex liquid) extends up to much higher fields[3,4] $H_{peak}(T) \sim H_{c2}(T)$, where $H_{peak}(T)$ is the position of the peak in $\rho_{xx}(H)$. For $T \rightarrow 0$, $H_{c2} \sim 20$ T for La$_{1.7}$Eu$_{0.2}$Sr$_{0.1}$CuO$_4$ and $H_{c2} \sim 25$ T for La$_{1.48}$Nd$_{0.4}$Sr$_{0.12}$CuO$_4$.

Figure 2 shows the field dependence of $R_H = \rho_{yx}(H)/H$ for various $T$ in both materials (see Supplementary Figs. 1 and 2 for the $\rho_{yx}(H)$ data at different $T$). At relatively high $T > T_0 > T_c^0$ in the pseudogap regime, the positive $R_H$ is independent of $H$ (Fig. 2a, c), as observed in conventional metals, although the in-plane transport is already insulatinglike, i.e. $d\rho_{xx}/dT < 0$ (Fig. 3a, c, also Supplementary Fig. 5). Upon cooling, $R_H$ decreases to zero at $T = T_0(H)$, and then becomes negative in the regime of lower fields. The field dependence remains weak at all $T$, similar to the observations[27] in striped La$_{1.905}$Ba$_{0.095}$CuO$_4$, but in contrast to the strong $H$-dependence of $R_H$ in YBa$_2$Cu$_3$O$_{6+x}$ and YBa$_2$Cu$_4$O$_8$ (YBCO materials; ref. [14]), i.e., in the absence of spin order, or in La$_{2-x}$Sr$_x$CuO$_4$ (ref. [25]), where the charge order is at best very weak. The most striking finding is that, at the

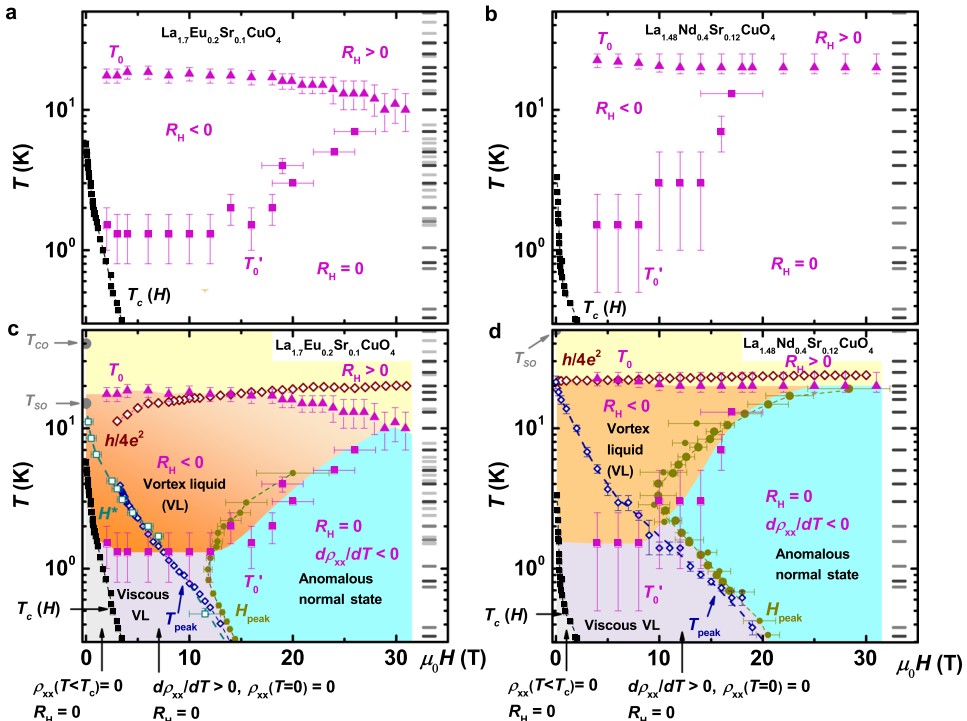

**Fig. 1 In-plane Hall coefficient $R_H$ across the $T$–$H$ phase diagram of striped cuprates. a, b** Regions of $T$ and $H$ with different signs of $R_H$ for La$_{1.7}$Eu$_{0.2}$Sr$_{0.1}$CuO$_4$ and La$_{1.48}$Nd$_{0.4}$Sr$_{0.12}$CuO$_4$, respectively. **c, d** Comparison of the results for $R_H$ to the other transport data[3,4] for La$_{1.7}$Eu$_{0.2}$Sr$_{0.1}$CuO$_4$ and La$_{1.48}$Nd$_{0.4}$Sr$_{0.12}$CuO$_4$, respectively. $T_c(H)$ (black squares): boundary of the vortex solid in which $\rho_{xx}(T < T_c) = 0$ and $R_H = 0$, as expected for a superconductor. The upper critical field $H_{c2}(T) \sim H_{peak}(T)$; $H_{peak}(T)$ (dark green dots) are the fields above which the magnetoresistance changes from positive to negative[3,4]. The low-$T$, viscous vortex liquid (VL) regime (light violet) is bounded by $T_c(H)$ and, approximately, by $T_{peak}(H)$ (positions of the peak in $\rho_{xx}(T)$; open blue diamonds), $H^*(T)$ (crossover between non-Ohmic and Ohmic behavior[3]; open royal squares), or $H_{peak}(T)$; here the behavior is metallic ($d\rho_{xx}/dT > 0$) with $\rho_{xx}(T \to 0) = 0$ and $R_H = 0$. The field-revealed normal state (blue) exhibits anomalous behavior: $\rho_{xx}(T)$ has an insulating, $\ln(1/T)$ dependence[3,4], but $R_H = 0$ despite the absence of superconductivity. At high $T$ (yellow), $R_H > 0$ and drops to zero at $T = T_0(H)$ (magenta triangles). In the high-$T$ VL regime ($H < H_{peak}$; dark beige), $R_H$ becomes negative before vanishing at lower $T = T_0'(H)$ (magenta squares), as the vortices become less mobile. The $h/4e^2$ symbols (open brown diamonds) show the ($T$, $H$) values where the sheet resistance changes from $R_{\square/layer} < R_Q = h/4e^2$ at higher $T$, to $R_{\square/layer} > R_Q$ at lower $T$. Zero-field values of $T_{SO}$ and $T_{CO}$ are also shown; $T_{PG} \sim 175$ K and $\sim 150$ K for La$_{1.7}$Eu$_{0.2}$Sr$_{0.1}$CuO$_4$ and La$_{1.48}$Nd$_{0.4}$Sr$_{0.12}$CuO$_4$, respectively[55]. All dashed lines guide the eye. In all panels, gray horizontal marks indicate measurement temperatures in different runs, the resolution of which defines vertical error bars for $T_0$ and $T_0'$; horizontal error bars reflect the uncertainty in defining $T_0'$ within our experimental resolution (see Supplementary Fig. 3 for the raw $R_H(H)$ data).

**Table 1 Characteristic temperatures.**

| Sample | $T_c^0$ (K) | $T_{SO}$ (K) | $T_{CO}$ (K) | $T_{PG}$ (K) |
|---|---|---|---|---|
| La$_{1.7}$Eu$_{0.2}$Sr$_{0.1}$CuO$_4$ | (5.7 ± 0.3) | ~ 15 (ref. [54]) | ~ 40 (ref. [54]) | ~ 175 (ref. [55]) |
| La$_{1.48}$Nd$_{0.4}$Sr$_{0.12}$CuO$_4$ | (3.6 ± 0.4) | ~ 50 (ref. [56]) | ~ 70 (ref. [56]) | ~ 150 (ref. [55]) |

highest fields ($H > H_{peak} \sim H_{c2}$), $R_H$ remains immeasurably small for $T < T_0$, down to the lowest measured $T$ (Fig. 2b, d). In other words, for a fixed $T < T_0(H)$, $R_H < 0$ at low $H$, but it becomes zero and remains zero (see Methods) with increasing field.

In Fig. 3, we compare $R_H(T)$ and $\rho_{xx}(T)$ for various fields. The drop of $R_H$ observed at $T > T_0$ does not depend on $H$ (Fig. 3b, d), similar to earlier studies of the striped La-214 family[27–31] and other cuprates[21]. The independence of the drop of $R_H$ on field implies that this is a property of the zero-field state, as opposed to some field-induced phase. In YBCO, the drop in $R_H$ was attributed[18,21] to the Fermi surface reconstruction by charge order. In striped cuprates, however, the onset of the drop in $R_H$ seems closer to the structural phase transition temperature $T_{d2}$ (Fig. 3), where $T_{SO} < T_{CO} < T_{d2} < T_{PG}$ (ref. [3]), but its origin is still under debate[27–31]. We define $T_0(H)$ as the temperature at which $R_H$ becomes zero or negative, and it is apparent that it has a very

weak, almost negligible field dependence. $T_0$ [$\sim (2 - 3)T_c^0$ for La$_{1.7}$Eu$_{0.2}$Sr$_{0.1}$CuO$_4$; $\sim 6T_c^0$ for La$_{1.48}$Nd$_{0.4}$Sr$_{0.12}$CuO$_4$] is comparable to the temperature at which $\rho_{xx}(T)$ curves in both materials split into either metalliclike (i.e. SClike) or insulatinglike, a correlation that seems to be manifested only in the presence of stripes[31]. We find that, interestingly, this occurs (Fig. 3a, c) when the normal state sheet resistance $R_{\square/layer} \approx R_Q$, where $R_Q = h/(2e)^2$ is the quantum resistance for Cooper pairs.

**Transport in the high-$T$, $H < H_{c2}$ regime.** Previous studies have identified[3,4] the $H < H_{peak}$ regime as the vortex liquid. The Hall resistivity due to mobile vortex cores is expected[32,33] to obey the relation $\rho_{xx}^2/\rho_{yx} \propto H$, which is indeed observed in this regime in our samples (Supplementary Fig. 6), thus confirming its identification as the vortex liquid. We also find that, in this field range,

 3

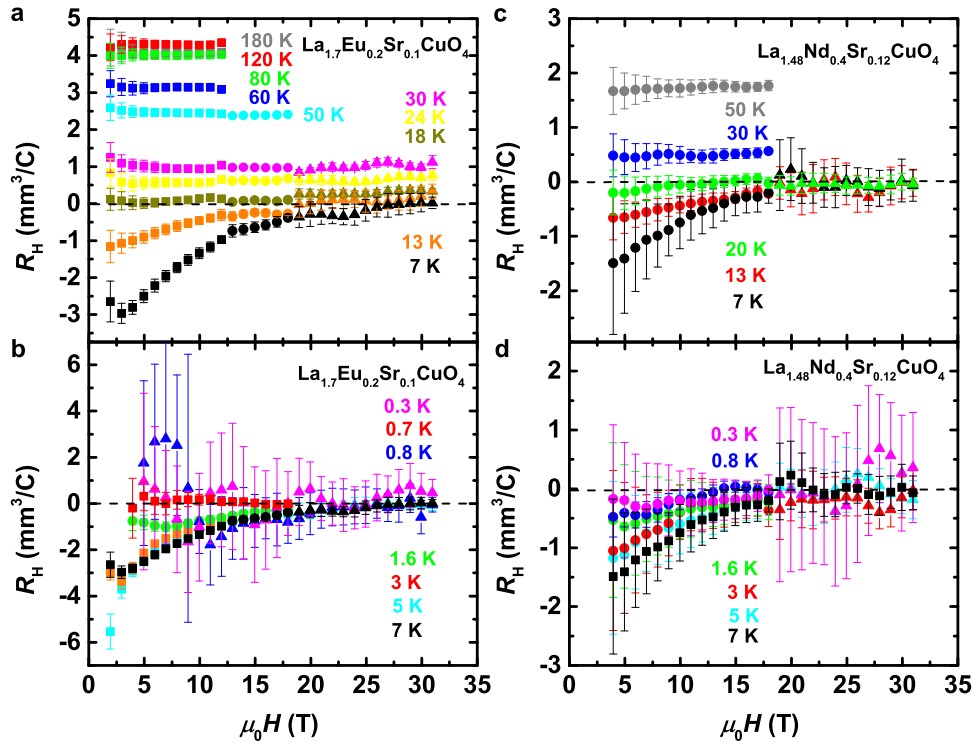

**Fig. 2 Field dependence of the Hall coefficient $R_H$ at various temperatures.** Higher- and lower-$T$ data for La$_{1.7}$Eu$_{0.2}$Sr$_{0.1}$CuO$_4$ are shown in **a** and **b**, respectively, i.e., in **c** and **d** for La$_{1.48}$Nd$_{0.4}$Sr$_{0.12}$CuO$_4$. Different symbols, corresponding to the data taken in different magnet systems, show good agreement between the runs. The data points represent $R_H$ values averaged over 1 T bins (Supplementary Fig. 3), while error bars correspond to ±1 SD (standard deviation) of the data points within each bin. The error bars are typically larger at lower $T$ resulting from the use of lower excitation currents $I$ (see Methods) necessary to avoid heating and to ensure that the measurements are taken in the $I \rightarrow 0$ limit, because of the strongly nonlinear (i.e., non-Ohmic) transport in the presence of vortices[3]. At higher $T$, the error bars are 3-4 times smaller, $\Delta R_H \sim 0.2 - 0.3$ mm$^3$/C (see also Supplementary Fig. 3). However, similar $\Delta R_H$, and even $\Delta R_H \sim 0.05$ mm$^3$/C, have been achieved also at low $T$, as described in Methods (see also Supplementary Fig. 4). At high $T$, $R_H$ is independent of $H$, but it decreases to zero at $T = T_0(H)$ upon cooling. As $T$ is reduced further, $R_H$ becomes negative for lower $H$, within the VL regime [$H < H_{c2}(T)$]. In the normal state [$H > H_{c2}(T)$], however, $R_H \approx 0$ down to the lowest $T$; $\Delta R_H \sim 0.05$ mm$^3$/C.

$R_H$ is negative for $T_0' < T < T_0$ (dark beige areas in Fig. 1c, d) and it exhibits a minimum, which is suppressed by increasing $H$ (Fig. 3b, d). Such behavior is generally understood[14,30,31] to result from the vortex contribution to $\rho_{yx}$. The minimum is less pronounced for $x \approx 1/8$ (Fig. 3d) than for $x = 0.10$ (Fig. 3b), consistent with prior observations[30,31], as well as with the recent evidence[4] of a more robust SC PDW state at $x \approx 1/8$.

Therefore, the agreement of the results of different techniques allows an unambiguous interpretation of the negative $R_H$ as being dominated by the motion of vortices, even if other effects might, in principle, also contribute to $R_H$. For example, in contrast to stripe-ordered La-214, in YBa$_2$Cu$_3$O$_y$ the negative $R_H$ increases with increasing $H$ (ref. [14]), suggesting that other effects dominate over the vortex contribution. Our results, however, show that the observation of a field-independent $T_0$, at which $R_H$ changes sign, does not necessarily imply that $R_H < 0$ is not caused by vortices.

**Transport in the low-$T$, $H < H_{c2}$ regime.** Similarly, at lower temperatures for $H < H_{peak}$, in the viscous VL region[3], the negative $R_H$ is suppressed by decreasing $T$, resulting in $R_H = 0$ at $T < T_0'$ (light violet area in Fig. 1c, d) down to the lowest measured $T = 0.019$ K (Fig. 3b, d insets). Here, $R_H = 0$ is thus attributed to the slowing down and freezing of the vortex motion with decreasing $T$ in the presence of disorder. This observation is reminiscent of the zero Hall resistivity observed within the VL regime ($H < H_{c2}$) in some conventional disordered 2D superconductors[34,35] and oxide interfaces[36]. Indeed, it has been

proposed[33] that the vanishing of $R_H$ in such so-called "failed superconductors" can be also explained by the strong pinning of the vortex motion.

Incidentally, our results (Fig. 1 and Supplementary Fig. 5) clarify that the origin of $R_H = 0$ observed earlier[30] in La$_{1.48}$Nd$_{0.4}$Sr$_{0.12}$CuO$_4$ for $T \lesssim 5$ K at 9 T is due to the onset of freezing of the vortex motion. Recently, $R_H = 0$ was reported[37] also in La$_{2-x}$Ba$_x$CuO$_4$ with $x = 1/8$, in the regime of nonlinear (i.e., non-Ohmic) transport analogous to the VL in Fig. 1, in which the negative $R_H$ arising from the vortex motion decreases towards zero as the doping approaches $x = 1/8$ (Fig. 3b, d). The vanishing Hall response in La$_{1.875}$Ba$_{0.125}$CuO$_4$ was indeed attributed[37–39] to the presence of SC phase fluctuations and Cooper pairs that survive within the charge stripes after the inter-stripe SC phase coherence has been destroyed by $H$. Likewise, in YBa$_2$Cu$_3$O$_y$ thin films near a disorder-tuned superconductor-insulator transition, $R_H = 0$ was found[40] below the onset $T$ (~80 K) for SC fluctuations, at low fields up to 9 T and in the regime of strong positive MR consistent with the suppression of superconductivity. Both refs. [37,40] reported $R_H = 0$ in an anomalous metallic regime with $\rho_{xx}(T \rightarrow 0) \neq 0$, similar to "failed superconductors" (or Bose metals)[34,35]. However, we note that in contrast, and unlike "failed superconductors"[34,35], in La$_{1.7}$Eu$_{0.2}$Sr$_{0.1}$CuO$_4$ and La$_{1.48}$Nd$_{0.4}$Sr$_{0.12}$CuO$_4$, as in highly underdoped La$_{2-x}$Sr$_x$CuO$_4$ (ref. [41]), $\rho_{xx}(T \rightarrow 0) = 0$ in the viscous VL[3,4]. In any case, we conclude that, in the entire $H < H_{c2}$ regime of these stripe-ordered cuprates, the Hall response is dominated by vortex physics.

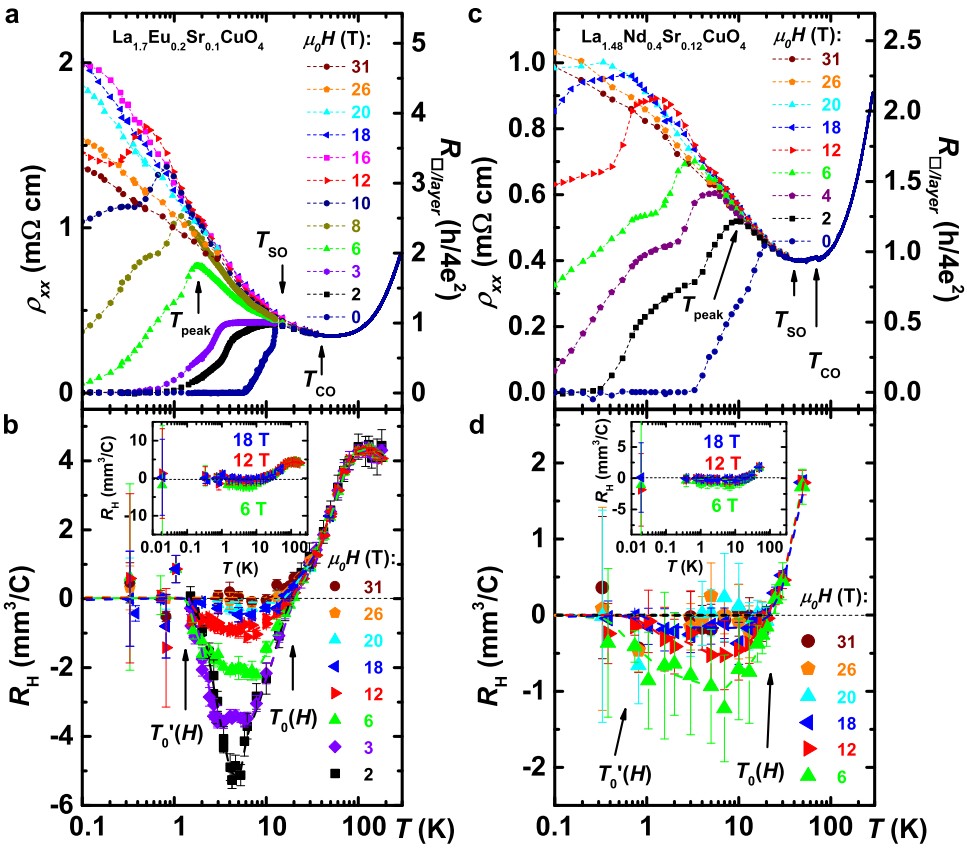

**Fig. 3 Temperature dependence of the in-plane longitudinal resistivity $\rho_{xx}$ and the Hall coefficient $R_H$ for various perpendicular $H$. a**, **b** $\rho_{xx}$ and $R_H$, respectively, for La$_{1.7}$Eu$_{0.2}$Sr$_{0.1}$CuO$_4$; the pseudogap temperature $T_{PG} \sim 175$ K (ref. [55]). **c**, **d** $\rho_{xx}$ and $R_H$, respectively, for La$_{1.48}$Nd$_{0.4}$Sr$_{0.12}$CuO$_4$; $T_{PG} \sim 150$ K (ref. [55]). The transition from the low-temperature orthorhombic to a low-temperature tetragonal structure occurs at $T_{d2} \sim 125$ K in La$_{1.7}$Eu$_{0.2}$Sr$_{0.1}$CuO$_4$ and $T_{d2} \sim 70$ K in La$_{1.48}$Nd$_{0.4}$Sr$_{0.12}$CuO$_4$ (ref. [3]). The data in **a** and **c** are from refs. [3,4]. At the highest fields, $\rho_{xx} \propto \ln(1/T)$, as discussed in more detail elsewhere[3]. In both materials, $R_H$ decreases upon cooling, and reaches zero at $T = T_0(H)$. For $H < H_{c2} \sim H_{peak}$, $R_H$ becomes negative at even lower $T$, then goes through a minimum, and eventually reaches zero again at $T = T_0'(H)$, as shown; $R_H$ remains zero down to 0.019 K (**b** and **d** insets). For $H > H_{c2}$, $R_H = 0$ for all $H$ and $T < T_0(H)$. Similar to those in Fig. 2, error bars correspond to ±1 SD of the data points within each bin. All dashed lines guide the eye.

**Transport in the $H > H_{c2}$ regime.** The remaining, most intriguing question is the origin of $R_H = 0$ observed beyond the VL regime, at all $T < T_0$ and $H > H_{peak} \approx H_{c2}$ (blue areas in Fig. 1c, d). This anomalous normal state is also characterized[3] by $\rho_{xx} \propto \ln(1/T)$. In addition, here the out-of-plane resistivity has the same $T$-dependence[4], $\rho_c \propto \ln(1/T)$, implying that the transport mechanism is the same for both in-plane and $c$ directions. We discuss several potential scenarios for the origin of $R_H = 0$ in this regime.

**Discussion**

For $H > H_{c2}$, the first possibility to consider is whether there are any remnants of superconductivity, such as SC fluctuations that may no longer be detectable in the $\rho_{xx}$ measurement. In cuprates (ref. [23] and refs. therein), as well as in conventional superconductors[23,42], the effect of SC fluctuations on the Hall signal has been extensively studied in the high-$T$ normal state, at low fields and above $T_c^0$, within the conventional, weak-pairing fluctuation formalism built upon the Ginzburg-Landau (GL) theory of the BCS regime. The qualitative picture of SC fluctuations at low temperatures and high fields ($H > H_{c2}$), however, drastically differs from the GL one[23,43], but in either case, existing models predict nonzero $R_H$ with particle-hole asymmetry terms[23,43]. Recently, a strong-pairing fluctuation theory that also incorporates pseudogap effects has been proposed[24] for $R_H$ in cuprates, but only for the low-field, $T > T_c^0$ regime. However, it does

not describe the $H$-independence of the drop in $R_H$ with decreasing $T$ observed for $T > T_0$ (Fig. 3b, d).

Extensive transport studies[3,4], including those of the anisotropy ratio $\rho_c/\rho_{xx}$, have not found any observable signs of superconductivity, including the PDW, for $H > H_{c2}$. For example, here $\rho_c/\rho_{xx}$ no longer depends on a magnetic field, neither $H\|c$ nor $H\perp c$, and it reaches its high-$T$, normal-state value. As discussed elsewhere[3], the value of $H_{c2} \approx H_{peak}$ is also consistent with the spectroscopic data for the closing of the SC gap in other cuprates. Although other experiments might be needed to definitively rule out the presence of any preformed pairs at $H > H_{peak}$, it appears far more likely that pairs cannot be responsible for $R_H = 0$ in the field-revealed normal state of La$_{1.7}$Eu$_{0.2}$Sr$_{0.1}$CuO$_4$ and La$_{1.48}$Nd$_{0.4}$Sr$_{0.12}$CuO$_4$, given also that $R_H = 0$ spans a $\sim$10 T-wide range of fields in Fig. 1. Therefore, models that rely on the existence of preformed pairs[38,39], strong SC correlations such as those in "failed superconductors"[34,35], or conventional Gaussian SC fluctuations[23,43] do not seem relevant for the $H > H_{c2}$ regime. Hence, we consider other possible scenarios.

The drop of the positive $R_H(T)$ to zero, observed at $T > T_0$, has been attributed[14,19,21,22] to the Fermi surface reconstruction, implying the presence of both hole and electron pockets in the Fermi surface. Although this issue is not fully settled[44], partly because of the disagreement with photoemission experiments, a similar drop of $R_H(T)$ seen in La$_{1.7}$Eu$_{0.2}$Sr$_{0.1}$CuO$_4$ and La$_{1.48}$Nd$_{0.4}$Sr$_{0.12}$CuO$_4$ at $T > T_0$ (Fig. 3b, d) suggests the possibility that the same mechanism might be responsible for the normal-state behavior of $R_H$ at $T > T_0$ in

these stripe-ordered cuprates, and even in their $T < T_0$, high-field regime, which is the focus of our study. We note, however, that there is no consensus on how the Fermi surface is affected by the presence of spin stripes, including in La$_{2−x−y}$Sr$_x$(Nd,Eu)$_y$CuO$_4$ compounds near $x = 1/8$. Therefore, without additional input from other techniques, any multiband model with a sufficient number of fitting parameters could reproduce our result that, in the high-field normal state, $R_H = 0$ within our measurement resolution (1 SD), $\Delta R_H \sim$ 0.05 mm$^3$/C (or standard error ~0.01 mm$^3$/C; see Methods). We note that the latter is comparable to, if not better than, $\Delta R_H$ in other similar studies (see Methods for a detailed discussion of the experimental resolution and unique measurement challenges). However, our results, in fact, place stringent constraints on any realistic models for the Hall effect in this regime: $R_H < 0.05$ mm$^3$/C, but this condition also needs to be satisfied over a wide range of $H$ and $T$ for two different materials and doping levels (Fig. 1). In a multiband picture, this would require that a subtle balance, or a near-cancellation, of contributions from hole and electron pockets is maintained over a huge range of parameters $T$ and $H$, as well as change in $x$ and the rare-earth composition $y$. Therefore, a multiband picture seems unlikely considering the robustness of our results.

Since $d\rho_{xx}/dT < 0$ in the normal state, one could speculate whether $R_H$ vanishes (i.e. $\rho_{yx} = 0$, or conductivity $\sigma_{xy} = 0$) because of some kind of localization. Strong, exponential localization does not describe the data because the $T$-dependence of the resistivity is very weak, it becomes even weaker with increasing $H$, and at the same time, the absolute value of $\rho_{xx}$ remains relatively low and comparable to that at $T > T_0$ (Fig. 3a, c). Similarly, as the system goes from the VL to the normal state with $H$ at a fixed, relatively high $T < T_0$, the $H$-dependence of $\rho_{xx}$ is negligible[3] (e.g., at ~ 4 K in Fig. 3a), while $R_H$ changes qualitatively from a finite negative value to zero (Fig. 3b). Our results for $R_H$ are indeed the opposite of those in lightly-doped[17], i.e. insulating cuprates with a diverging $\rho_{xx}(T \to 0)$, or in highly underdoped[16] cuprates, both of which seem to show a diverging $R_H$ at low $T$. If $n = n_H = 1/(eR_H)$ holds, this is indeed consistent with a depletion of carriers, whereas in our case it would indicate a diverging number of carriers. Likewise, weak localization in 2D is not consistent with the data, since the same $\ln(1/T)$ behavior is observed also along the $c$ axis, just like in underdoped La$_{2−x}$Sr$_x$CuO$_4$ (ref. [6]). While weak localization does not produce a correction to the classical $R_H$ value, electron-electron interactions in weakly disordered 2D metals give rise[45] to logarithmic corrections to $\rho_{xx}$ and $R_H$, which are related such that $\delta R_H/R_H = 2(\delta\rho_{xx}/\rho_{xx})$. However, just like in La$_{2−x}$Sr$_x$CuO$_4$ (ref. [6]), this is not consistent with our observation[3] of a large $\ln(1/T)$ term in $\rho_{xx}$, and it does not describe the vanishing $R_H$. Hence, standard localization mechanisms cannot explain $R_H = 0$ observed over a wide range of $T < T_0$ and $H > H_{c2}$.

On the other hand, a confinement of carriers within 1D charge stripes, associated with the suppression of the cyclotron motion with increasing $H$, was proposed to understand the drop of the positive $R_H(T)$ towards zero observed[28] in La$_{2−x−y}$Nd$_y$Sr$_x$CuO$_4$ near $x = 1/8$ at low $H = 5$ T and high $T$, i.e. $T > T_0$ in Fig. 3b, d. Although, in contrast, our central result is $R_H = 0$ in the high-field ($H > H_{c2}$), $T < T_0$ regime, models based on the quasi-1D picture seem to be a plausible description of stripe-ordered cuprates also when the applied $H$ suppresses the interstripe Josephson coupling. One such model, for example, predicts[46], both in the presence and the absence of a spin gap, a non-Fermi-liquid smectic metal phase, in which the transport across the stripes is incoherent, whereas it is coherent inside each stripe. Importantly, a smectic metal has an approximate particle-hole symmetry[46] for $x < 1/8$, which implies $\rho_{yx} \approx 0$, as observed in our experiment. Incidentally, the same model had been proposed as the origin of the drop of $R_H$ in the early studies[47] of YBa$_2$Cu$_3$O$_y$ at $T > T_0$. Other, more general scenarios include holographic models

for doped Mott insulators[48], which also feature emergent particle-hole symmetry[49].

Our study of the Hall effect across the entire in-plane $T$-$H$ phase diagram has clarified and further confirmed that the origin of $R_H = 0$ reported in earlier studies[28,30,31,37] of stripe-ordered cuprates is associated with the presence of SC fluctuations. In contrast, our central result is that, at much higher fields, such that $H > H_{c2}$, the field-revealed normal state of La$_{2−x−y}$Sr$_x$(Nd, Eu)$_y$CuO$_4$ cuprates with static spin and charge stripes is characterized by a zero, i.e. immeasurably small, Hall coefficient. Indeed, since the vanishing of $R_H$ is pronounced over a wider range of $H$ and $T$ for $x = 0.12$ (Fig. 1b, d) than for $x = 0.10$ (Fig. 1a, c), this strongly suggests that $R_H \approx 0$ is crucially related to the presence of static stripe order. Further insight into this issue might come from other experiments at high fields, such as optical conductivity, Raman scattering, and thermal transport, to determine whether $R_H \approx 0$ results from a fortuitous near-cancellation of contributions from multiple bands or it signals an approximate particle-hole symmetry, as expected for a smectic metal in a stripe-ordered cuprate[46] and in more general models of correlated matter[48,49].

## Methods

**Samples.** Several single crystal samples of La$_{1.8−x}$Eu$_{0.2}$Sr$_x$CuO$_4$ with a nominal $x = 0.10$ and La$_{1.6−x}$Nd$_{0.4}$Sr$_x$CuO$_4$ with a nominal $x = 0.12$ were grown by the traveling-solvent floating-zone technique[50]. The high quality of the crystals was confirmed by several techniques, as discussed in detail elsewhere[3,4]. The samples were shaped as rectangular bars suitable for direct measurements of the longitudinal and transverse (Hall) resistance, $R_{xx}$ and $R_{yx}$, respectively. Detailed measurements of $R_{xx}$ and $R_{yx}$ were performed on La$_{1.7}$Eu$_{0.2}$Sr$_{0.1}$CuO$_4$ sample "B" with dimensions $3.06 \times 0.53 \times 0.37$ mm$^3$ ($a \times b \times c$, i.e. length × width × thickness) and a La$_{1.48}$Nd$_{0.4}$Sr$_{0.12}$CuO$_4$ crystal with dimensions $3.82 \times 1.19 \times 0.49$ mm$^3$. The same two samples were also studied previously[3,4]. After ~ 3 years, the low-$T$ properties of the La$_{1.7}$Eu$_{0.2}$Sr$_{0.1}$CuO$_4$ sample "B" changed, which was attributed to a small change (increase) in the effective doping, but its phases remained qualitatively the same[4]. We repeated the Hall measurements after the sample had changed, and obtained the same results.

Gold contacts were evaporated on polished crystal surfaces, and annealed in air at 700 °C. The current contacts were made by covering the whole area of the two opposing sides with gold to ensure uniform current flow, and the voltage contacts were made narrow to minimize the uncertainty in the absolute values of the resistance. Multiple voltage contacts on opposite sides of the crystals were prepared, and the results did not depend on the position of the contacts. Gold leads (≈25 μm thick) were attached to the samples using the Dupont 6838 silver paste, followed by the heat treatment at 450 °C in the flow of oxygen for 15 min. The resulting contact resistances were less than 0.1 Ω for La$_{1.7}$Eu$_{0.2}$Sr$_{0.1}$CuO$_4$ (0.5 Ω for La$_{1.48}$Nd$_{0.4}$Sr$_{0.12}$CuO$_4$) at room temperature. Meanwhile, we found no change in the superconducting properties of the samples before and after the annealing.

**Measurements.** The standard ac lock-in techniques (~13 Hz) were used for measurements of $R_{xx}$ and $R_{yx}$ with the magnetic field parallel and anti-parallel to the $c$ axis. The Hall resistance was determined from the transverse voltage by extracting the component antisymmetric in the magnetic field. The Hall coefficient $R_H = R_{yx} d/H = \rho_{yx}/H$, where $d$ is the sample thickness. The $\rho_{xx}$ data measured simultaneously with $\rho_{yx}$ agree well with the previously reported results of magnetoresistance measurements[3,4]. The resistance per square per CuO$_2$ layer $R_{\square/\text{layer}} = \rho_{xx}/l$, where $l = 6.6$ Å is the thickness of each layer.

Depending on the temperature, the excitation current (density) of 10 μA to 316 μA (~$5 \times 10^{−3}$ A cm$^{−2}$ to ~ $1.6 \times 10^{−1}$ A cm$^{−2}$ for La$_{1.7}$Eu$_{0.2}$Sr$_{0.1}$CuO$_4$, and ~$2 \times 10^{−3}$ A cm$^{−2}$ to ~ $6.3 \times 10^{−2}$ A cm$^{−2}$ for La$_{1.48}$Nd$_{0.4}$Sr$_{0.12}$CuO$_4$) was used: 10 μA for 0.019 K (Supplementary Fig. 1d); 100 μA for all measurements in fields up to 12 T (Supplementary Fig. 1a), and for the 0.3 K data in Supplementary Figs. 2 and 3; 316 μA for all other measurements. These excitation currents were low enough to avoid Joule heating[3]. Traces with different excitation currents were also compared to ensure that the reported results are in the linear response regime. A 1 kΩ resistor in series with a $\pi$ filter [5 dB (60 dB) noise reduction at 10 MHz (1 GHz)] was placed in each wire at the room temperature end of the cryostat to reduce the noise and heating by radiation in all measurements.

Several different cryostats at the National High Magnetic Field Laboratory were used, including a dilution refrigerator (0.016 K ⩽ T ⩽ 0.7 K) and a $^3$He system (0.3 K ⩽ T ⩽ 35 K) in superconducting magnets ($H$ up to 18 T), using 0.1–0.2 T/min sweep rates, and a $^3$He system (0.3 K ⩽ T ⩽ 20 K) in a 31 T resistive magnet, using 1–2 T/min sweep rates. Some of the measurements were performed in a variable-temperature insert (1.7 K ≤ T ≤ 200 K) with a 12 T superconducting magnet. The fields were swept at constant temperatures, and the sweep rates were

low enough to avoid eddy current heating of the samples. The results obtained in different magnets and cryostats agree well.

**Experimental resolution of Hall effect measurements.** Unlike other cuprates such as YBCO, in which $\rho_{xx}$ and $\rho_{yx}$ are comparable at low $T$ and high $H$ (e.g. ref. [51]), $\rho_{yx}$ is orders of magnitude smaller than $\rho_{xx}$ in La$_{1.7}$Eu$_{0.2}$Sr$_{0.1}$CuO$_4$ and La$_{1.48}$Nd$_{0.4}$Sr$_{0.12}$CuO$_4$, even in the high-$T$ normal state. For example, as seen from Supplementary Fig. 1a for La$_{1.7}$Eu$_{0.2}$Sr$_{0.1}$CuO$_4$ at $T = 180$ K and $H = 12$ T, $\rho_{yx} = R_{yx}d \sim 0.005$ mΩ cm, while $\rho_{xx} \gtrsim 0.5$ mΩ cm (see Fig. 3 for $H = 0$, but at $T > 15$ K, the magnetoresistance is very weak[3,52]). At low $T$, $\rho_{yx}$ is drastically suppressed even further (Supplementary Fig. 1), and the ratio $\rho_{yx}/\rho_{xx}$ becomes even greater. This observation is significant in itself as discussed in the main text, but it also presents certain experimental challenges.

In a standard Hall measurement, any contribution of $\rho_{xx}$, which results from a slight misalignment of voltage contacts, is removed and $\rho_{yx}$ is isolated by antisymmetrization of the transverse voltage drops measured with a field both parallel and antiparallel to the $c$ axis. However, a perfect cancellation of the $\rho_{xx}$ contribution can only be achieved if the two measurements in opposite field directions are conducted at exactly the same $T$ (and other experimental conditions). Otherwise, $\rho_{xx}$ can contaminate the Hall resistivity even after the conventional antisymmetrization procedure, especially if $\rho_{xx}$ is much larger than $\rho_{yx}$ and it has a strong temperature dependence as in La$_{1.7}$Eu$_{0.2}$Sr$_{0.1}$CuO$_4$ and La$_{1.48}$Nd$_{0.4}$Sr$_{0.12}$CuO$_4$. Therefore, careful temperature control during the experiment and meticulous data analysis afterwards are key to our Hall measurements on these two systems.

With the single-shot $^3$He cryostat, the temperature control below 1.6 K is usually complicated by the evaporation of the $^3$He liquid, which induces a slow $T$ drift with time. To minimize its impact, we measured the traces with opposite fields in back-to-back experiments and did not consider the data when the $T$ drift was too large. The maximum $T$ drift between the two traces is typically $\sim 10 - 20$ mK for $T < 1.6$ K.

To ensure the accuracy of our results, we have also repeated Hall measurements on La$_{1.7}$Eu$_{0.2}$Sr$_{0.1}$CuO$_4$ at 0.71 K more than 10 times, by recondensing the $^3$He liquid and resetting the temperature for each positive and negative field sweep. This ensures that, even if $T$ drifts due to $^3$He evaporation, the amount of the drift would be the same in the two traces. We carefully compared the field dependence of the Cernox® thermometer reading, $T_r$, for each positive and negative field sweep, a typical example of which is shown in Supplementary Fig. 4a inset. We note that the Cernox® sensor is not calibrated in the field, and thus the increase of $T_r$ only reflects the magnetoresistance of the sensor, while the sample temperature (controlled by the sorb) is unchanged. As shown in the Supplementary Fig. 4a inset, the temperature is the same (within 1 mK) during the entire positive and negative field sweeps.

To determine the uncertainty of the Hall coefficient measurement results, we divide the $R_{yx}(H)$ and $R_H(H)$ data into bins (typical size is 1 T; see Supplementary Figs. 2 and 3), and calculate the mean and the standard deviation (SD) within each bin. Therefore, the error bars in Figs. 2 and 3, and Supplementary Figs. 1, 2, 3, 4, and 6, all correspond to $\pm 1$ SD of the data points within each bin. To reduce the SD even further, we averaged over five sets of measurements at 0.71 K to reduce the experimental error bar (i.e. 1 SD) from $\Delta R_H \sim 0.2$ mm$^3$/C to $\Delta R_H \sim 0.05$ mm$^3$/C (Supplementary Fig. 4b). This is comparable to, if not better than, $\Delta R_H$ in other studies of the Hall effect on cuprates[14], including those in which zero Hall coefficient (induced by superconductivity, not in the normal state) was found[27,30,37,40], as well as on other systems, such as iron-based superconductors[53]. We emphasize again that the experimental error for $\rho_{yx}$ (and $R_H$) is dominated by the imperfect cancellation of the contribution from the $T$-dependent longitudinal resistivity $\rho_{xx}$, which is inevitably much larger than the (nearly) zero transverse contribution. At $T = 0.71$ K, where we achieved almost perfect temperature control (to within 1 mK) and thus the maximum cancellation of the longitudinal resistivity contribution (Supplementary Fig. 4a), we also determined, using standard error analysis, the $\sim 95\%$ confidence intervals for $R_H$, e.g., $-0.008 \pm 0.020$ mm$^3$/C at 17 T. This further confirms our conclusion that the Hall coefficient (and Hall resistivity, see Supplementary Fig. 1e inset) remains zero in the high-field normal state, i.e., above the upper critical field $H_{peak}$.

In principle, the error bar in the $\rho_{yx}$ measurement can also be reduced by increasing the excitation current density or, equivalently, by reducing the sample thickness or width for a fixed current. However, the applied current density still needs to remain below the limit above which Joule heating is induced. The effects of excitation currents have been studied thoroughly[3], so that here we have used the highest excitation current density possible without inducing Joule heating. Therefore, reducing the sample thickness, for example, would not help to decrease the error bar further, because a smaller excitation current would also need to be used.

## Data availability

The data that support the findings of this study are available within the paper and the Supplementary Information. Additional data related to this paper may be requested from the authors.

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

## Acknowledgements

We thank G. Saraswat for experimental assistance, and J. M. Tranquada, K. Yang, J. Zaanen for helpful discussions. This work was supported by NSF Grants Nos. DMR-1307075 and DMR-1707785, and the National High Magnetic Field Laboratory (NHMFL) through the NSF Cooperative Agreements Nos. DMR-1157490, DMR-1644779, and the State of Florida. Z.S. acknowledges support by the Priority Academic Program Development of Jiangsu Higher Education Institutions (PAPD). Z.S. is also grateful for the support from Jiangsu Key Laboratory of Thin Films and Jiangsu Key Lab of Advanced Optical Manufacturing Technologies.

## Author contributions

Single crystals were grown and prepared by T.S.; Z.S., P.G.B., J.T. and B.K.P. performed the measurements; Z.S. analyzed the data; Z.S. and D.P. wrote the manuscript, with input from all authors; D.P. supervised the project.

## Competing interests

The authors declare no competing interests.
