## [Peer Review File · Nature Communications]

REVIEWER COMMENTS

Reviewer #1 (Remarks to the Author):

Shi and co-workers examine the electronic ground state of two LNSCO and LESCO cuprate superconductors and observe an unusual experimental signature: zero Hall resistivity over a range of magnetic fields above H_{c2} . This novel observation follows on the heels of prior studies of that have observed zero Hall response in other cuprate materials, as well as very active investigations of the vanishing Hall response proximal to a true superconducting state in disordered, quasi-2D superconductors. The authors ultimately link their observations to the stripe-ordered phase, and it is likely that future experimental and theoretical work will develop a complete picture of the underlying physics.

The authors' data are comprehensive and represent a significant accomplishment; Hall effect studies to such high fields well below 1 K are technically demanding. The manuscript is clearly written and organized; given the depth of the cuprate field and the range of behaviors that the study touches on, the paper will (perhaps necessarily) be inaccessible to many non-specialist readers. The measurements follow recent studies of nonlinear and anisotropic transport in Refs. 11 and 12 and builds off of the work of Ref. 11 in particular, which also hints at a "anomalous normal state" at high magnetic fields beyond the vortex regime. Although the central experimental finding of the work is clear, the specific implications for the physics of LSCO or related cuprate families are not. The authors' text concludes with the suggestion that this is related to stripe-phase behavior, but a microscopic physical picture or mechanism would be particularly useful to readers' physical insight. Similarly, without systematic trends with doping it is unclear how extended the observation will prove to be; the fact that the data appear from a single sample for each material further limits the breadth of implications for understanding cuprate physics.

There are several questions / comments whose answers could strengthen the clarity and/or conclusions of the work.

1) The authors do not quantitatively examine any microscopic existing models for the Hall effect (beyond the vortex liquid regime); there are many for both the cuprates (e.g. Phys. Rev. B 99, 134504) and for contributions due to SC fluctuations (e.g. Phys. Rev. B 86, 014515) that may be relevant. (I am not an author on these papers, or any mentioned below; I found them after a cursory search, and would expect that further review may expand the utility of the manuscript for readers.)

2) Following on the previous point, readers would benefit from explicit discussion of the relevant physics in the high-field zero-Hall-effect phase, particularly since fluctuations at short length scales may persist to high fields. Are there conclusive regions to reject the “failed superconductor” picture (reviewed comprehensively in Rev. Mod. Phys. 91, 011002)?

3) Finally, are there non-phenomenological approaches to estimating the size of the Hall resistance (at comparable fields) in phases that could then be excluded from contention? This could strengthen the significance of the $R_H = 0$ finding; at present it is unclear if there is any ‘reasonable’ explanation for a small (but nonzero) Hall contribution that would also be consistent with the authors’ data. For example, recent studies have tracked fluctuation Hall contributions to σ_{xy} in low- T_c superconductors (see e.g. Phys. Rev. B 95, 224501 and its references), while phenomenological studies in cuprates have an extensive literature that could be more referenced/quantitatively contrasted with the current study.

4) In the discussion of the $H > H_{c2}$ regime, the authors state that $\rho_{xy} = 0$ could equal zero “because of some kind of quasiparticle localization”, without reference to a picture or model – it is not clear whether they mean something distinct from e.g. the WL/WAL $\ln(T)$ corrections known to show up in R_{xy} (see e.g. Phys. Rev. B 22, 5142 and citing work).

5) The caption of several figures mentions “1 SD of the data points within each bin”. Since the relevant uncertainty estimator for the mean of a collection of points in a “bin” is the standard error, this language is confusing and may mean that the authors may need to adjust their uncertainties, or adopt an alternate phrasing (such as “the $\pm 1\sigma$ error bars were calculated using the SEM”). Since the authors are quoting results that are supposedly “consistent with zero”, clarifying this point is central to evaluating the manuscript. (Similarly, phrases such as “within error” in the text (p4, p7) promulgate the imprecise statement that a datapoint’s error bar must “overlap with zero” to be consistent.)

7) The main text (p3) states $\rho_{xy} = R_H * H$; assuming a right-handed coordinate system the convention of positive R_H for holes means that $R_H = \rho_{yx}/H$.

8) There are a few other technical points that may clarify details for readers:

- Several of the figures use bright red/green color contrast that will be impossible to differentiate for a significant fraction of readers. Figure 1, critical to the authors’ story, is so complex as to almost be unreadable; showing the “data” and “schematic phase diagram” plots side-by-side might improve the clarity / impact to readers.

- It appears that some fraction fraction of the data in e.g. Figure S3 is re-plotted from the authors' Ref 11; readers may benefit from clarity in the caption about which elements are reproduced from the earlier work.

- The notation " $T = T_0(H) \sim (2 - 6) T_c$ " in e.g. p4 is ambiguous.

- The phrase "weak insulatinglike" is not specific; stating that " dR/dT is negative" and " $R \sim \ln(T)$ " (and/or associating these statements with the novel term) would be more informative to readers.

Reviewer #2 (Remarks to the Author):

The present paper by Shi et. al investigates the longitudinal (ρ_{xx}) and transverse (ρ_{xy}) resistivities in LaSr(Nd,Eu)CuO superconductors over a wide range of both temperature and magnetic field values. The Hall response (ρ_{xy}) is an important characteristic since it depends on the sign of the constituent charge carriers and vanishes in the presence of particle-hole symmetry. One of the most intriguing findings in this paper is the vanishing of the Hall coefficient (R_H) in the field-revealed normal state, as indicated in the blue region in Fig. 1. This observed behaviour of R_H is strikingly different from the properties of say YBCO, which, as shown in Ref. 19, tends to plateau in the pseudogap phase for low temperatures and high fields. Similarly, the behaviour observed here is different from the LaSrCuO studies in Ref. 2, which observed a plateau in R_H for large magnetic fields and decreasing temperatures.

Another one of the authors' key results is that, in the high-field limit, as temperature decreases R_H decreases from a positive value down to zero at temperatures $T_0 \sim (2 - 6)T_c$, and it remains zero down to the lowest measured temperatures. The weak magnetic field dependence of T_0 leads the authors to conclude that $R_H \approx 0$ is a characteristic of the zero-field (normal) ground state in the presence of stripes. Emery et. al proposed that a suppression in R_H could be synonymous with stripe physics and an effective particle-hole symmetry. Based on this idea, here the authors have suggested that, because the behaviour of the high-field Hall coefficient is fundamentally different from other cuprates, there may be an approximate particle-hole symmetry that is unique to stripe-ordered cuprates.

Possible suppression in R_H due to multiband physics, such as simultaneous electron and hole carriers, is convincingly argued by the authors to not be a possible explanation of the experimental

results. Such an explanation would need to be robust over a wide range of temperatures, magnetic fields, and rare-earth composition, which, as argued, seems unlikely. A similar argument against multiband physics explaining a sign change in $R_{\{H\}}$ was put forth in the case of YBCO studies by Jin and Ott in PRB 57, 13872 (1998). The authors also suggest that preformed pairs are not responsible for the vanishing of $R_{\{H\}}$, given this result spans a wide range of magnetic field values. This also seems consonant with high-field weak-fluctuation Cooper-pairs literature [RMP 90, 015009 (2018)], which has not observed such phenomena. Thus, there are compelling arguments to attribute, at least suggestively, the observed phenomenon to particle-hole symmetry and stripe phenomena.

The authors have discussed in depth the vortex liquid phase part of the diagram, and overall the entire phase diagram and its boundaries have been meticulously analysed. The high-field normal state that has been investigated (blue region in Fig. 1) is an exciting regime for physicists to consider.

Future investigation of the thermal Hall response would provide an additional probe of the apparent particle-hole symmetry in the anomalous normal state since κ_{xy} , like σ_{xy} , is nonzero only when this symmetry is broken.

Overall I am in favour of the publication of this article. I have some minor comments listed below that would be helpful to clarify.

(i) It would be useful to give $T_c(H=0)$, $T_{\{CO\}}$, and $T_{\{SO\}}$ for each sample when discussing Fig. 1; that is, include the paragraph on page 20 of the methods section in the main text, to provide comparative scales. In Fig. 1(a) the temperature range of the observed anomalous normal state does not seem to extend up to $\sim 6T_c(H=0)$. For Fig. 1(b), however, the temperature range does indeed seem to extend to $6T_c(H=0)$. I think it would be better to give the temperature range for each sample independently, if not in the abstract then certainly in the main text.

(ii) It would also be helpful to note what $T^*(H=0)$ (the pseudogap temperature) is for each sample.

(iii) On page 7 the authors mention the drops in $R_{\{H\}}$ observed in YBCO superconductors, and they state that this was attributed to the properties of the zero-field pseudogap regime. However, as I understand Ref. 21, and more recently Ref. 19, the observed decreases in $R_{\{H\}}$ in YBCO superconductors were for dopings below the onset of the pseudogap phase and in Ref. 19 they were attributed to CDW phenomena.

(iv) I would also like to point out that Ando and Segawa in J. Phys. Chem. Solids 63, 2253 (2002) observed a drop in $R_{\{H\}}$ in a YBCO compound, and they speculated that this drop could be due to particle-hole symmetric stripe phenomena. Granted they did not observe a wide range of temperatures where $R_{\{H\}}=0$, as the present paper does, nevertheless the speculative reason for the observed results was based on the idea of Emery et. al (Ref. 34), which is also being suggested here as an explanation for the results. The authors may find this work to be relevant to cite.

Reviewer #3 (Remarks to the Author):

This manuscript reports on magnetotransport measurements in the cuprate superconductor samples exhibiting stripe order. The authors find that the Hall resistance vanishes in otherwise metallic compounds. The most significant new result is that Hall resistance remains zero at magnetic field values that are substantially higher than the upper critical field, where finite Hall resistance is expected for a conventional metal. This significant experimental result will help narrow down theoretical models of high-Tc cuprates. I believe that the study is of substantial interest to the broader research community and is suitable for publication in Nature Communications, after some minor edits.

Questions to authors:

The authors state that they confirm that the origin of $R_H = 0$ is associated with the presence of SC fluctuations. How does this mechanism extend above H_{c2} where SC is suppressed?

Fig 1 combines the newly measured Hall effect and previously published transport measurements. Are these and prior measurements conducted on the same two samples? Is there a variation in the properties of the samples with the same nominal composition?

Are samples' superconducting properties affected by contact annealing at 700 C and 450 C?

What are the actual values for the current, at least for the low T traces on Extended Figs. 1,2, and 6?

Reply to the Reviewers

We are grateful to the reviewers for reading our manuscript and for their constructive suggestions on content and clarity. We welcome the opportunity to improve our manuscript. In the revised manuscript, we have addressed all of the reviewers' comments. The changes are summarized below (comments in blue, replies to comments in black). Unless noted otherwise, the reference numbers correspond to those in the current version of the manuscript.

Reviewer #1 (Remarks to the Author):

Shi and co-workers examine the electronic ground state of two LNSCO and LESCO cuprate superconductors and observe an unusual experimental signature: zero Hall resistivity over a range of magnetic fields above H_{c2} . This novel observation follows on the heels of prior studies of that have observed zero Hall response in other cuprate materials, as well as very active investigations of the vanishing Hall response proximal to a true superconducting state in disordered, quasi-2D superconductors. The authors ultimately link their observations to the stripe-ordered phase, and it is likely that future experimental and theoretical work will develop a complete picture of the underlying physics.

The authors' data are comprehensive and represent a significant accomplishment; Hall effect studies to such high fields well below 1 K are technically demanding. The manuscript is clearly written and organized; given the depth of the cuprate field and the range of behaviors that the study touches on, the paper will (perhaps necessarily) be inaccessible to many non-specialist readers. The measurements follow recent studies of nonlinear and anisotropic transport in Refs. 11 and 12 and builds off of the work of Ref. 11 in particular, which also hints at a "anomalous normal state" at high magnetic fields beyond the vortex regime. Although the central experimental finding of the work is clear, the specific implications for the physics of LSCO or related cuprate families are not. The authors' text concludes with the suggestion that this is related to stripe-phase behavior, but a microscopic physical picture or mechanism would be particularly useful to readers' physical insight. Similarly, without systematic trends with doping it is unclear how extended the observation will prove to be; the fact that the data appear from a single sample for each material further limits the breadth of implications for understanding cuprate physics.

We thank the reviewer for the overall positive assessment of our accomplishments and for appreciating the significance of our work. We also agree with the reviewer that expanding the discussion about possible microscopic mechanisms underlying our central finding and putting it into a broader context of research on cuprates and other superconductors would strengthen our manuscript further. This is indeed what we have done in the revised manuscript, as described in detail below in response to the reviewer's specific suggestions. Just to illustrate the scope of the expanded discussion, here we note that we have added 12 new references to the paper.

Regarding the samples, we note that our Hall measurements are a part of a larger project on transport properties of striped cuprates. Some results have already been published (refs. 3 and 4), as also noted by the reviewer. As shown in refs. 3 and 4, several single crystals of each of the two striped cuprates have been studied, and we have found strikingly consistent results using multiple transport techniques. Our detailed Hall measurements on two typical samples of the two different striped cuprate materials and two different doping levels further support the phase diagrams we established previously and the robustness of our conclusions.

As described in our response to one of the comments of reviewer 3, we also performed Hall measurements on LESCO sample “B1” with an effective doping that was intermediate (ref. 4) to those of the two samples presented here, and we obtained the same result. We have added a few sentences in Methods, section “Samples”, where this is discussed. We respectfully refer the reviewer to our response to reviewer 3’s comment #2 for more details.

Therefore, our measurements reveal a robust, intrinsic property of striped cuprates at doping levels near $x=1/8$ (i.e. $x\sim 0.10-0.12$) in the $T/T_{c0} \ll 1$ and $H > H_{c2}$ regime that has never been explored before, partly because of the significant technical challenges as also acknowledged by the reviewer. Since our central finding, the vanishing of the Hall coefficient in the normal state over such an extended range of T and H is unprecedented in condensed matter science, we believe that our manuscript merits publication in Nature Communications. Furthermore, besides revealing this new fundamental property and highlighting the important role of competing orders that are present in all hole-doped cuprates, our paper has important implications for the interpretation of Hall measurements in other cuprate families, some of which are still controversial. In particular, we have expanded the discussion in which our findings in various parts of the (T, H) phase diagram are compared to those on YBCO and LSCO. Because of the breadth and complexity of the problem, expanding the study to dopings even further away from $1/8$ will be the focus of future research, and so will the study of other striped cuprates.

There are several questions / comments whose answers could strengthen the clarity and/or conclusions of the work.

1) The authors do not quantitatively examine any microscopic existing models for the Hall effect (beyond the vortex liquid regime); there are many for both the cuprates (e.g. Phys. Rev. B 99, 134504) and for contributions due to SC fluctuations (e.g. Phys. Rev. B 86, 014515) that may be relevant. (I am not an author on these papers, or any mentioned below; I found them after a cursory search, and would expect that further review may expand the utility of the manuscript for readers.)

We thank the reviewer for the helpful suggestion. As the reviewer noted, the depth of the cuprate field is tremendous after more than three decades of intensive research, and identifying the mechanisms underlying the results of numerous Hall measurements has been an everlasting and ongoing effort. However, the existing microscopic models focus mostly on a) superconducting (SC) fluctuations, and b) a change in the Hall number across the charge order and the pseudogap quantum critical points at much higher doping. In those papers, contributions to a *finite* Hall resistivity by different mechanisms are intensively discussed, but are not directly relevant to our main result, the *vanishing* of the Hall coefficient in a large phase space. The two

papers mentioned by the reviewer, for example, both focus on the contributions to the Hall effect from SC fluctuations and are not directly relevant to our findings.

Nevertheless, we completely agree with the reviewer that the manuscript would benefit from an expanded discussion of models that might have some potential relevance to our results. Therefore, we have made the following changes in the paper in response to the reviewer's comment.

(1) On p.3, paragraph 2, we have added:

“In general, the magnitude of R_H reflects the degree of particle-hole asymmetry and, thus, understanding the Hall coefficient provides deep insight into the microscopic properties.”

We have also modified the text that follows to expand its scope.

“However, the interpretation of the Hall effect in cuprates has been a challenge, because R_H can depend on both T and H , and it can be affected by various factors, such as the presence of SC correlations and the topological structure of the Fermi surface. For example, ...”

(2) On p. 4, paragraph 1, the following sentences have been added.

“Other studies of the Hall effect in cuprates have focused on the effects of SC fluctuations (refs. ^{23, 24} and refs. therein), and on the pronounced change in the Hall number across the charge order and the pseudogap quantum critical points^{16–18, 25, 26}. However, the Hall behavior in the $T \rightarrow 0$, $H > H_{c2}$ regime has remained mostly unexplored. In particular, ...”

Here, references 23-26 are new; Boyack et al., Phys. Rev. B 99, 134504 mentioned by the reviewer is now ref. 24.

(3) On p. 7, paragraph 2, we have added the following sentence and reference 36.

“The Hall resistivity due to mobile vortex cores is expected^{35, 36} to obey the relation $\rho_{xx}^2/\rho_{yx} \propto H$, which is indeed observed in this regime in our samples (Supplementary Fig. 6), thus confirming its identification as the vortex liquid.”

(4) At bottom of p. 8, subsection “Transport in the $H > H_{c2}$ regime”, we have added the following text, because it is relevant for the subsequent expanded discussion of possible models in the high-field region and to the reviewer's next comment.

“This anomalous normal state is also characterized³ by $\rho_{xx} \propto \ln(1/T)$. In addition, here the out-of-plane resistivity has the same T -dependence⁴, $\rho_c \propto \ln(1/T)$, implying that the transport mechanism is the same for both in-plane and c directions. We discuss several potential scenarios for the origin of $R_H=0$ in this regime.”

(5) On p. 9, at the beginning of the Discussion section, we have added the following paragraph, which is focused on SC fluctuations.

“For $H > H_{c2}$, the first possibility to consider is whether there are any remnants of superconductivity, such as SC fluctuations that may no longer be detectable in the ρ_{xx} measurement. In cuprates (ref.²³ and refs. therein), as well as in conventional superconductors^{23, 45}, the effect of SC fluctuations on the Hall signal has been extensively studied in the high- T normal state, at low fields and above T_c^0 , within the conventional, weak-pairing fluctuation formalism built upon the Ginzburg-Landau (GL) theory of the BCS regime. The qualitative picture of SC fluctuations at low temperatures and high fields ($H > H_{c2}$), however, drastically differs from the GL one^{23, 46}, but in either case, existing models predict nonzero R_H with particle-hole asymmetry terms^{23, 46}. Recently, a strong-pairing fluctuation theory that also incorporates pseudogap effects has been proposed²⁴ for R_H in cuprates, but only for the low-field, $T > T_c^0$ regime. However, it does not describe the H -independence of the drop in R_H with decreasing T observed for $T > T_0$ (Fig. 3b,d).”

Here, references 45 and 46 are also new; Michaeli et al., Phys. Rev. B 86, 014515 mentioned by the reviewer is ref. 46.

(6) We have added the following text at top of p. 10, in the Discussion section, which focuses on the $H > H_{c2}$ regime. (Please see also our response to the reviewer’s next comment.)

“Therefore, models that rely on the existence of preformed pairs^{41, 42}, strong SC correlations such as those in “failed superconductors”^{37, 38}, or conventional Gaussian SC fluctuations^{23, 46} do not seem relevant for the $H > H_{c2}$ regime.”

Here, refs. 38 and 42 are new; ref. 38 was mentioned by the reviewer in the next comment.

(7) The paragraph on multiband effects was moved up to p. 10 to improve the readability of the manuscript.

Other changes to the manuscript that discuss possible models or microscopic mechanisms are described in response to the reviewer’s comments below. Some additional, related changes have been made in response to the comments of reviewers #2 and #3.

2) Following on the previous point, readers would benefit from explicit discussion of the relevant physics in the high-field zero-Hall-effect phase, particularly since fluctuations at short length scales may persist to high fields. Are there conclusive regions to reject the “failed superconductor” picture (reviewed comprehensively in Rev. Mod. Phys. 91, 011002)?

We had indeed carefully considered the “failed superconductor” scenario in our manuscript (pp. 7-8, subsection “Transport in the low- T , $H < H_{c2}$ regime.”). Nevertheless, the reviewer’s comment prompts us to expand this discussion further. In particular, we have split that subsection into two paragraphs for clarity, and added the following text and the new ref. 39 to the end of the first paragraph, right after “ R_H is thus attributed to the slowing down and freezing of the vortex motion with decreasing T ...” at bottom of p. 7.

“...in the presence of disorder. This observation is reminiscent of the zero Hall resistivity observed within the VL regime ($H < H_{c2}$) in some conventional disordered 2D superconductors^{37, 38} and oxide interfaces³⁹. Indeed, it has been proposed³⁶ that the vanishing of R_H in such so-called “failed superconductors” can be also explained by the strong pinning of the vortex motion.”

The second paragraph of that subsection includes our previous discussion of the experiments in $\text{YBa}_2\text{Cu}_3\text{O}_y$ thin films (ref. 43) and $\text{La}_{1.875}\text{Ba}_{0.125}\text{CuO}_4$ crystals (ref. 40), in which the vanishing R_H was interpreted within a “failed superconductor” picture. As we discussed, those observations seem to correspond to those in our low- T , $H < H_{c2}$ regime, where clear evidence of SC correlations has been found^{3, 4}. We also highlighted some differences, by adding “and unlike “failed superconductors”^{37, 38}” near the end of that paragraph. We note again that all previous experimental papers report the vanishing of R_H *within* the VL regime, in contrast to the key finding of our study.

The reviewer’s question is about whether the same picture can be applied to our $H > H_{c2}$ regime. The entire section “Discussion” of our paper, starting on p. 9, is devoted to the discussion of possible mechanisms that might give rise to $R_H=0$ in the high-field normal state. A “failed superconductor” should feature significant superconducting correlations, however we have not found any signs of superconductivity for $H > H_{c2}$. In particular, in our previous studies^{3, 4} we determined H_{c2} using *four* different approaches, all of which agree very well. These include high- T Gaussian fluctuation fitting, both linear and nonlinear transport at low T , and anisotropy between the out-of-plane and the in-plane resistivity. Those results were further supported by energy scale considerations based on reports of spectroscopy studies. Therefore, above H_{c2} in our phase diagram we find absolutely no observable signs of superconductivity, including the PDW, which implies that the “failed superconductor” picture is highly unlikely to describe the vanishing Hall response in the large phase space above H_{c2} , spanning a 10 T-wide range of fields. This is all described on p. 9 (second paragraph) and top of p. 10 of our manuscript.

Since this comment is related to the reviewer’s comment 1), some of the relevant changes are described above, in particular change (6).

We also note that we discussed other possible mechanisms that might be relevant for the high-field region, such as the smectic metal model proposed by Emery et al. (ref. 49) on p. 12, which might be the best framework for explaining our results. We added one sentence to that part of the manuscript, following a suggestion of reviewer #2; please see our response to reviewer #2.

3) Finally, are there non-phenomenological approaches to estimating the size of the Hall resistance (at comparable fields) in phases that could then be excluded from contention? This could strengthen the significance of the $R_H = 0$ finding; at present it is unclear if there is any ‘reasonable’ explanation for a small (but nonzero) Hall contribution that would also be consistent with the authors’ data. For example, recent studies have tracked fluctuation Hall contributions to σ_{xy} in low- T_c superconductors (see e.g. Phys. Rev. B 95, 224501 and its references), while phenomenological studies in cuprates have an extensive literature that could be more referenced/quantitatively contrasted with the current study.

We agree with the reviewer that, ideally, it would be interesting and helpful to compare our results with some theoretical models. We have indeed carefully considered the origin of $R_H=0$, and also discussed it with many experts, both theorists and experimentalists. As mentioned above and described in the revised manuscript, most of the literature on the Hall effect is not relevant for our findings in the $H > H_{c2}$ regime. The reference that the reviewer mentioned is an example of studies of contributions of Gaussian SC fluctuations to the Hall resistivity in NbN films; it focuses on the regime of $T > T_c^0$, where thermal fluctuations dominate.

The Gaussian theory in the opposite, low- T and $H > H_{c2}$ limit is very different (refs. 23, 46), and it has been tested, for example, on disordered TaN films [Brezney et al., PRB 86, 014504 (2012)], i.e. in another conventional superconductor. However, it is not obvious whether such theory is, first of all, relevant for cuprates: it has been argued (e.g. ref. 24) that strong-pairing fluctuation theory would be more appropriate and that pseudogap effects need to be taken into account. In addition, the effects of charge and spin orders on the Fermi surface would need to be taken into account, but there is no theory that includes all of these effects. Even if we ignored these caveats and wished to compare our data, i.e. the upper limit on R_H , to the results of standard approaches (refs. 23, 46), it would be impossible to estimate the value of the particle-hole asymmetry factor that features in the formula because the information about the Fermi surface in these stripe-ordered cuprates is insufficient and controversial (please see p.10, top of paragraph 2 of our manuscript). This illustrates further the challenges of interpreting Hall effect (and many other!) measurements in cuprates. Here we also remark that even the in-plane and out-of-plane resistivities exhibit anomalous behavior in the $H > H_{c2}$ regime that cannot be described using standard theoretical approaches, and hence it is not necessarily surprising that standard theories may not explain the vanishing of R_H in such an anomalous normal state.

We have discussed this issue in our response to the reviewer's previous comments, where we also described how we have expanded the relevant discussions in the revised manuscript. The changes are summarized in our response to those comments above.

4) In the discussion of the $H > H_{c2}$ regime, the authors state that $\rho_{xy} = 0$ could equal zero "because of some kind of quasiparticle localization", without reference to a picture or model – it is not clear whether they mean something distinct from e.g. the WL/WAL $\ln(T)$ corrections known to show up in R_{xy} (see e.g. Phys. Rev. B 22, 5142 and citing work).

We thank the reviewer for pointing out the part of our discussion that was insufficiently clear. In the revised paper, for clarity we have expanded this paragraph and discussed both strong and weak localization mechanisms, as well as the effects of electron-electron interactions in weakly disordered 2D metals. We have added ref. 48, a well-known review article that discusses the paper mentioned by the reviewer and other related work. The revised text (second paragraph on p. 11) reads as follows.

"Since $d\rho_{xx}/dT < 0$ in the normal state, one could speculate whether R_H vanishes (i.e. $\rho_{yx}=0$, or conductivity $\sigma_{xy}=0$) because of some kind of localization. Strong, exponential localization does not describe the data because the T -dependence of the resistivity is very weak, it becomes even weaker with increasing H , and at the same time, the absolute value of ρ_{xx} remains relatively low and comparable to that at $T > T_0$ (Fig. 3a,c). Similarly, as the system goes from the VL to the

normal state with H at a fixed, relatively high $T < T_0$, the H -dependence of ρ_{xx} is negligible³ (e.g. at ~ 4 K in Fig. 3a), while R_H changes qualitatively from a finite negative value to zero (Fig. 3b). Our results for R_H are indeed the opposite of those in lightly-doped¹⁷, i.e. insulating cuprates with a diverging ρ_{xx} ($T \rightarrow 0$), or in highly underdoped¹⁶ cuprates, both of which seem to show a diverging R_H at low T . If $n = n_H = 1/(eR_H)$ holds, this is indeed consistent with a depletion of carriers, whereas in our case it would indicate a diverging number of carriers. Likewise, weak localization in 2D is not consistent with the data, since the same $\ln(1/T)$ behavior is observed also along the c axis, just like in underdoped $\text{La}_{2-x}\text{Sr}_x\text{CuO}_4$ (ref. 6). While weak localization does not produce a correction to the classical R_H value, electron-electron interactions in weakly disordered 2D metals give rise⁴⁸ to logarithmic corrections to ρ_{xx} and R_H , which are related such that $\delta R_H/R_H = 2\delta\rho_{xx}/\rho_{xx}$. However, just like in $\text{La}_{2-x}\text{Sr}_x\text{CuO}_4$ (ref. 6), this is not consistent with our observation³ of a large $\ln(1/T)$ term in ρ_{xx} , and it does not describe the vanishing R_H . Hence, standard localization mechanisms cannot explain $R_H=0$ observed over a wide range of $T < T_0$ and $H > H_{c2}$.”

5) The caption of several figures mentions “1 SD of the data points within each bin”. Since the relevant uncertainty estimator for the mean of a collection of points in a “bin” is the standard error, this language is confusing and may mean that the authors may need to adjust their uncertainties, or adopt an alternate phrasing (such as “the $\pm 1\sigma$ error bars were calculated using the SEM”). Since the authors are quoting results that supposedly “consistent with zero”, clarifying this point is central to evaluating the manuscript. (Similarly, phrases such as “within error” in the text (p4, p7) promulgate the imprecise statement that a datapoint’s error bar must “overlap with zero” to be consistent.)

We agree with the reviewer about the need to make precise statements. Therefore, we have made several changes in the manuscript in response to the reviewer’s comment.

First, the phrases “within error” and “within experimental error”, now on pages 4 and 6, have been replaced by “(see Methods)” to refer the reader to the expanded discussion in Methods for details.

The standard deviation (SD) measures the dispersion of individual data points relative to its mean, while the standard error measures the likely discrepancy between the calculated average of the data points and the true mean of the population. The error bars for R_H that we show in the paper were calculated as the SD of the collection of data points within each bin, and they are not the standard error of the mean. In the revised manuscript, we have defined the error bars clearly, and retained SD as the measure of the uncertainty in order to facilitate the comparison to the literature where SD is typically provided. However, in Methods, we have also added the values that we obtained from the standard error analysis. The following changes were made.

(1) Fig. 2 caption: “...error bars correspond to ± 1 SD (standard deviation) of the data points within each bin.”

(2) Fig. 3 caption: “Similar to those in Fig. 2, error bars correspond to ± 1 SD of the data points within each bin.”

(3) In Methods, (new) paragraph 3 on p. 16, under section “Experimental resolution of Hall effect measurements”, we have added the following.

“To determine the uncertainty of the Hall coefficient measurement results, we divide the $R_{yx}(H)$ and $R_H(H)$ data into bins (typical size is 1 T, see Supplementary Figs. 2 and 3), and calculate the mean and the standard deviation (SD) within each bin. Therefore, the error bars in Figs. 2 and 3, and Supplementary Figs. 1, 2, 3, 4, and 6, all correspond to ± 1 SD of the data points within each bin. To reduce the SD even further...”

In the middle of the same paragraph (top of p. 17), we have added:

“We emphasize again that the experimental error for ρ_{yx} (and R_H) is dominated by the imperfect cancellation of the contribution from the T -dependent longitudinal resistivity ρ_{xx} , which is inevitably much larger than the (nearly) zero transverse contribution. At $T=0.71$ K, where we achieved almost perfect temperature control (to within 1 mK) and thus the maximum cancellation of the longitudinal resistivity contribution (Supplementary Fig. 4a), we also determined, using standard error analysis, the $\sim 95\%$ confidence intervals for R_H , e.g. (-0.008 ± 0.020) mm^3/C at 17 T. This further confirms...”

(4) On p. 10, we have further clarified the definition of the experimental resolution:

“... our measurement resolution (1 SD), $\Delta R_H \sim 0.05 \text{ mm}^3/\text{C}$ (or standard error $\sim 0.01 \text{ mm}^3/\text{C}$; see Methods).”

7) The main text (p3) states $\rho_{xy} = R_H * H$; assuming a right-handed coordinate system the convention of positive R_H for holes means that $R_H = \rho_{yx}/H$.

We thank the reviewer for this comment [we note that there is no comment (6)]. We have made the correction and replaced ρ_{xy} and R_{xy} with ρ_{yx} and R_{yx} respectively throughout the main text and supplementary, including those in Supplementary Figs. 1, 2, and 6, and their captions.

8) There are a few other technical points that may clarify details for readers:

- Several of the figures use bright red/green color contrast that will be impossible to differentiate for a significant fraction of readers.

We thank the reviewer for reminding us regarding the color scheme choice. We have modified the colors in Fig. 1 and avoided bright red/green and blue/green color combinations whenever possible. We have also made sure that different quantities are represented by different symbol shapes, not just different colors. We have modified the corresponding text throughout the paper.

Figure 1, critical to the authors’ story, is so complex as to almost be unreadable; showing the “data” and “schematic phase diagram” plots side-by-side might improve the clarity / impact to readers.

We thank the reviewer for the comment and for the helpful suggestion. Indeed, Fig. 1 is rather complex as it contains data from different types of measurements including those reported in our previous publications. We note that it is with these comprehensive experimental results that we were able to establish a complete T - H phase diagram for the two striped cuprates. However, we do realize that it could be difficult for the readers to follow, and have made the following changes motivated by the reviewer's suggestion.

We have modified Fig. 1, and added two new panels as (a) and (b). The new panels show only the "data", i.e. the phase diagrams or regions of T and H defined only by the Hall data [and $T_c(H)$]. The phase diagrams in previous Fig. 1a,b now appear as Fig. 1c,d. In this way, the regions with different signs of the Hall coefficient and their relationship with the different phases are now much clearer.

We have modified the main text accordingly, and added the following on p. 5, paragraph 2.

"Our main results are shown in Fig. 1. From the Hall measurements, we are able to identify regions in (T,H) phase space with different signs of R_H (Fig. 1a,b) and, in particular, we find $R_H \approx 0$ over a wide range of T and H in both materials. Further insight is obtained by comparing the Hall results with the phase diagram obtained by other transport techniques, as shown in Fig. 1c,d."

- It appears that some fraction of the data in e.g. Figure S3 is re-plotted from the authors' Ref 11; readers may benefit from clarity in the caption about which elements are reproduced from the earlier work.

We thank the reviewer for the comment. We believe the reviewer meant Fig. 3 instead of Fig. S3. In response to the reviewer's comment, we have modified Fig. 3 caption to clarify the origin of the data.

"The data in **a** and **c** are from refs. ^{3,4}."

- The notation " $T = T_0(H) \sim (2 - 6) T_c$ " in e.g. p4 is ambiguous.

We thank the reviewer for pointing out the ambiguity in our notation. Indeed, reviewer 2 also made a similar comment. As a response, we have made the following changes.

(1) On p. 5, we have removed the phrase " $(2 - 6) T_c$ ", and added a sentence specifying the $T_0(H)$ scales for the two materials.

"Here, $T_0(H) \sim (2-3)T_c$ for $\text{La}_{1.7}\text{Eu}_{0.2}\text{Sr}_{0.1}\text{CuO}_4$ and $T_0(H) \sim 6T_c$ for $\text{La}_{1.48}\text{Nd}_{0.4}\text{Sr}_{0.12}\text{CuO}_4$;"

(2) On p. 6, we have replaced the phrase " $(2 - 6) T_c$ " with the following.

" T_0 [$\sim (2-3)T_c$ for $\text{La}_{1.7}\text{Eu}_{0.2}\text{Sr}_{0.1}\text{CuO}_4$; $\sim 6T_c$ for $\text{La}_{1.48}\text{Nd}_{0.4}\text{Sr}_{0.12}\text{CuO}_4$]"

We have kept the phrase in the abstract unchanged, since its implication is now explained in the main text, and we think it would have been too detailed to specify the temperature range for each material in the abstract.

- The phrase “weak insulatinglike” is not specific; stating that “ dR/dT is negative” and “ $R \sim \ln(T)$ ” (and/or associating these statements with the novel term) would be more informative to readers.

We thank the reviewer for pointing out the confusing terminology. In response to the reviewer’s comment, we have made the following changes to make the terminology consistent with that in the literature: $dR/dT < 0$ is equivalent to insulatinglike; $\rho_{xx} \propto \ln(1/T)$ is insulating, but weak in comparison to exponential temperature dependence. In particular:

(1) In the abstract, we have replaced “weakly insulatinglike behavior” with “weak insulating behavior of the resistivity, i.e. $\rho_{xx} \propto \ln(1/T)$, ...”.

We note that “ dR/dT is negative” is implied by “ $\rho_{xx} \propto \ln(1/T)$ ”.

(2) On p. 6, we have removed “weakly” because the emphasis of the text was on $d\rho_{xx}/dT < 0$, which is commonly synonymous with “insulatinglike”.

(3) On p. 3 and in Fig. 1 caption, “insulatinglike” was replaced by “insulating”, since it is defined there already as the $\ln(1/T)$ dependence.

We thank the reviewer for the constructive comments, which have helped us to expand our discussion and further strengthen our conclusions. The reviewer’s comments have also helped us to clarify and improve our presentation significantly. We hope the reviewer now finds our revised manuscript suitable for publication in Nature Communications.

Reviewer #2 (Remarks to the Author):

The present paper by Shi et. al investigates the longitudinal (ρ_{xx}) and transverse (ρ_{xy}) resistivities in LaSr(Nd,Eu)CuO superconductors over a wide range of both temperature and magnetic field values. The Hall response (ρ_{xy}) is an important characteristic since it depends on the sign of the constituent charge carriers and vanishes in the presence of particle-hole symmetry. One of the most intriguing findings in this paper is the vanishing of the Hall coefficient ($R_{\{H\}}$) in the field-revealed normal state, as indicated in the blue region in Fig. 1. This observed behaviour of $R_{\{H\}}$ is strikingly different from the properties of say YBCO, which, as shown in Ref. 19, tends to plateau in the pseudogap phase for low temperatures and high fields. Similarly, the behaviour observed here is different from the LaSrCuO studies in Ref. 2, which observed a plateau in $R_{\{H\}}$ for large magnetic fields and decreasing temperatures.

Another one of the authors' key results is that, in the high-field limit, as temperature decreases $R_{\{H\}}$ decreases from a positive value down to zero at temperatures $T_0 \sim (2 - 6)T_c$, and it remains zero down to the lowest measured temperatures. The weak magnetic field dependence of T_0 leads the authors to conclude that $R_{\{H\}} \approx 0$ is a characteristic of the zero-field (normal) ground state in the presence of stripes. Emery et. al proposed that a suppression in $R_{\{H\}}$ could be synonymous with stripe physics and an effective particle-hole symmetry. Based on this idea, here the authors have suggested that, because the behaviour of the high-field Hall coefficient is

fundamentally different from other cuprates, there may be an approximate particle-hole symmetry that is unique to stripe-ordered cuprates.

Possible suppression in $R_{\{H\}}$ due to multiband physics, such as simultaneous electron and hole carriers, is convincingly argued by the authors to not be a possible explanation of the experimental results. Such an explanation would need to be robust over a wide range of temperatures, magnetic fields, and rare-earth composition, which, as argued, seems unlikely. A similar argument against multiband physics explaining a sign change in $R_{\{H\}}$ was put forth in the case of YBCO studies by Jin and Ott in PRB 57, 13872 (1998). The authors also suggest that preformed pairs are not responsible for the vanishing of $R_{\{H\}}$, given this result spans a wide range of magnetic field values. This also seems consonant with high-field weak-fluctuation Cooper-pairs literature [RMP 90, 015009 (2018)], which has not observed such phenomena. Thus, there are compelling arguments to attribute, at least suggestively, the observed phenomenon to particle-hole symmetry and stripe phenomena.

The authors have discussed in depth the vortex liquid phase part of the diagram, and overall the entire phase diagram and its boundaries have been meticulously analysed. The high-field normal state that has been investigated (blue region in Fig. 1) is an exciting regime for physicists to consider.

Future investigation of the thermal Hall response would provide an additional probe of the apparent particle-hole symmetry in the anomalous normal state since κ_{xy} , like σ_{xy} , is nonzero only when this symmetry is broken.

We appreciate the reviewer's insights and agree with his/her suggestion. Indeed, the thermal Hall response would be the natural next step. A giant thermal Hall conductivity has recently been observed in underdoped cuprates including striped cuprates at high temperatures and low fields [Grisonnanche et al, Nature 571, 376 (2019) and Nature Phys. 16, 1108 (2020)], but studies of the thermal Hall response in the low- T , high- H regime that is the focus of our paper are still not available. For example, thermal conductivity for $\text{La}_{1.48}\text{Nd}_{0.4}\text{Sr}_{0.12}\text{CuO}_4$ was reported by Michon et al., Phys. Rev. X 8, 041010 (2018), at temperatures 0.05-0.3 K, but only in fields up to 15 T – this corresponds to the viscous vortex liquid regime in Fig. 1 (also in refs. 3 and 4). We had indeed mentioned thermal transport as one of the promising techniques for further studies at the end of the manuscript; our work should motivate more studies in this regime.

Overall I am in favour of the publication of this article. I have some minor comments listed below that would be helpful to clarify.

We thank the reviewer for the careful reading of our manuscript and an accurate summarization of our results and their implications. We appreciate the recommendation for publication. Below we describe our response to the reviewer's comments.

(i) It would be useful to give $T_c(H=0)$, $T_{\{CO\}}$, and $T_{\{SO\}}$ for each sample when discussing Fig. 1; that is, include the paragraph on page 20 of the methods section in the main text, to provide comparative scales. In Fig. 1(a) the temperature range of the observed anomalous normal state does not seem to extend up to $\sim 6T_c(H=0)$. For Fig. 1(b), however, the temperature range

does indeed seem to extend to $6T_c(H=0)$. I think it would be better to give the temperature range for each sample independently, if not in the abstract then certainly in the main text.

We thank the reviewer for the helpful suggestions. In fact, reviewer 1 also made a similar comment. Therefore, we have made the following changes, also in response to the reviewer's next comment about the pseudogap temperature T_{PG} .

(1) We have moved the relevant paragraph from Methods to p. 5, where we have also added the values of T_{PG} and some related discussion, while specifying the $T_0(H)$ scales for each material.

“Here, $T_0(H) \sim (2-3)T_c^0$ for $\text{La}_{1.7}\text{Eu}_{0.2}\text{Sr}_{0.1}\text{CuO}_4$ and $T_0(H) \sim 6T_c^0$ for $\text{La}_{1.48}\text{Nd}_{0.4}\text{Sr}_{0.12}\text{CuO}_4$; T_c^0 , where the linear resistivity ρ_{xx} becomes zero, is (5.7 ± 0.3) K for $\text{La}_{1.7}\text{Eu}_{0.2}\text{Sr}_{0.1}\text{CuO}_4$ and (3.6 ± 0.4) K for $\text{La}_{1.48}\text{Nd}_{0.4}\text{Sr}_{0.12}\text{CuO}_4$. On the other hand, $T_{SO} \sim 15$ K, $T_{CO} \sim 40$ K, and the pseudogap temperature $T_{PG} \sim 175$ K ... Therefore, the vanishing Hall coefficient appears well below the pseudogap temperature, in the temperature region where both charge and spin orders (i.e. stripes) have fully developed. Meanwhile, we note that...”

(2) On p. 6, we have replaced the phrase “ $(2 - 6) T_c^0$ ” with the following, as discussed also in response to the reviewer 1's related comment.

“ $T_0 [\sim (2-3)T_c^0$ for $\text{La}_{1.7}\text{Eu}_{0.2}\text{Sr}_{0.1}\text{CuO}_4$; $\sim 6T_c^0$ for $\text{La}_{1.48}\text{Nd}_{0.4}\text{Sr}_{0.12}\text{CuO}_4$]

We have kept the phrase in the abstract unchanged, since its implication is now explained in the main text, and we think it would have been too detailed to specify the temperature range for each material in the abstract.

(ii) It would also be helpful to note what $T^*(H=0)$ (the pseudogap temperature) is for each sample.

We have added the pseudogap temperatures for both materials on p. 5, as described above. Since the pseudogap temperature is much higher than the scales shown in Figure 1, we have added this information in Figure 1 caption.

“ $T_{PG} \sim 175$ K and ~ 150 K for $\text{La}_{1.7}\text{Eu}_{0.2}\text{Sr}_{0.1}\text{CuO}_4$ and $\text{La}_{1.48}\text{Nd}_{0.4}\text{Sr}_{0.12}\text{CuO}_4$, respectively.”

(iii) On page 7 the authors mention the drops in $R_{\{H\}}$ observed in YBCO superconductors, and they state that this was attributed to the properties of the zero-field pseudogap regime. However, as I understand Ref. 21, and more recently Ref. 19, the observed decreases in $R_{\{H\}}$ in YBCO superconductors were for dopings below the onset of the pseudogap phase and in Ref. 19 they were attributed to CDW phenomena.

We thank the reviewer for pointing out the confusing statement. The reviewer is right that in YBCO the decrease of R_H happens for dopings below the CDW transition ($p \sim 0.16$), which is below the doping for the onset of the pseudogap ($p \sim 0.19$), as described in old ref. 19, now ref. 18. What we had in mind instead is the independence of the drop of R_H and T_0 on field (T_0 was defined in ref. 21 as the temperature where R_H drops below 0, i.e. the same as our definition).

This was pointed out in ref. 21, which then argued that “this independence of T_0 on field also shows that the modification of the Fermi surface implied by the sign change of R_H is characteristic of the zero-field pseudogap phase, not some field-induced ordered state”.

For clarity, we have modified the text on p. 6, paragraph 2, as follows.

“The independence of the drop of R_H on field implies that this is a property of the zero-field state, as opposed to some field-induced phase. In YBCO, the drop in R_H was attributed^{18,21} to the Fermi surface reconstruction by charge order. In striped cuprates, however, the onset of the drop in R_H seems closer to the structural phase transition temperature T_{d2} (Fig. 3), where $T_{SO} < T_{CO} < T_{d2} < T_{PG}$ (ref. ³), but its origin is still under debate^{30–34}.”

In addition, we have added the following sentence in Fig. 3 caption.

“The transition from the low-temperature orthorhombic to a low-temperature tetragonal structure occurs at $T_{d2} \sim 125$ K in $\text{La}_{1.7}\text{Eu}_{0.2}\text{Sr}_{0.1}\text{CuO}_4$ and $T_{d2} \sim 70$ K in $\text{La}_{1.48}\text{Nd}_{0.4}\text{Sr}_{0.12}\text{CuO}_4$ (ref. ³).”

(iv) I would also like to point out that Ando and Segawa in J. Phys. Chem. Solids 63, 2253 (2002) observed a drop in $R_{\{H\}}$ in a YBCO compound, and they speculated that this drop could be due to particle-hole symmetric stripe phenomena. Granted they did not observe a wide range of temperatures where $R_{\{H\}}=0$, as the present paper does, nevertheless the speculative reason for the observed results was based on the idea of Emery et. al (Ref. 34), which is also being suggested here as an explanation for the results. The authors may find this work to be relevant to cite.

We thank the reviewer for bringing this interesting paper to our attention. Indeed, the model proposed by Emery et al (ref. 49) was suggested early on to explain the drop of R_H in YBCO in the paper mentioned by the reviewer, before the effects of charge order in that compound became better understood in subsequent studies. Motivated by the reviewer’s comment, we have added the following sentence near the end of paragraph 1 on p. 12.

“Incidentally, the same model had been proposed as the origin of the drop of R_H in the early studies⁵⁰ of $\text{YBa}_2\text{Cu}_3\text{O}_y$ at $T > T_0$. Other, more general scenarios include ...”

In summary, we thank the reviewer for his/her insights and for the very constructive comments that we have used to improve our paper. We believe that our manuscript is now suitable for publication in Nature Communications.

Reviewer #3 (Remarks to the Author):

This manuscript reports on magnetotransport measurements in the cuprate superconductor samples exhibiting stripe order. The authors find that the Hall resistance vanishes in otherwise metallic compounds. The most significant new result is that Hall resistance remains zero at magnetic field values that are substantially higher than the upper critical field, where finite Hall resistance is expected for a conventional metal. This significant experimental result will help

narrow down theoretical models of high- T_c cuprates. I believe that the study is of substantial interest to the broader research community and is suitable for publication in Nature Communications, after some minor edits.

We are grateful to the reviewer for appreciating the significance of our results and for recommending its publication in Nature Communications.

Questions to authors:

The authors state that they confirm that the origin of $R_H = 0$ is associated with the presence of SC fluctuations. How does this mechanism extend above H_{c2} where SC is suppressed?

We thank the reviewer for bringing our attention to this potentially confusing part of our presentation. We have made the following changes to improve the clarity of our presentation and address the reviewer's comment. In fact, many of those changes were also prompted by the comments of reviewer 1.

a) First, we want to clarify that there are two regions in the phase diagram [the new Figs. 1(c) and 1(d)] where $R_H = 0$ is observed. There is a light violet region (labeled as "Viscous VL"), which is at fields below the upper critical field H_{c2} . Here, superconducting fluctuations (vortex liquid) dominate, and the zero Hall response is the result of superconductivity. On the other hand, there is another region (blue region, labeled as "Anomalous normal state") where $R_H = 0$, and this is the key result of our paper. Here, in the anomalous normal state, H is higher than H_{c2} and SC is completely suppressed. To show the phase diagram with different signs of R_H more clearly, we have added two panels in Fig. 1 (Fig. 1a,b), where only the results of the Hall experiments are shown, as mentioned in our response to reviewer 1's comment 8.

b) To clarify the discussion of different mechanisms responsible for the behavior of R_H in different parts of the phase diagram, in particular for $H < H_{c2}$ and $H > H_{c2}$, we have created a new subsection, "Transport in the $H > H_{c2}$ regime.", at the end of the Results section. This also addresses the first comment of reviewer 1, and further highlights our key result, which is the observation of $R_H = 0$ for $H > H_{c2}$.

"This anomalous normal state is also characterized³ by $\rho_{xx} \propto \ln(1/T)$. In addition, here the out-of-plane resistivity has the same T -dependence⁴, $\rho_c \propto \ln(1/T)$, implying that the transport mechanism is the same for both in-plane and c directions. We discuss several potential scenarios for the origin of $R_H = 0$ in this regime."

c) This is followed by the new section heading "Discussion", and new text "For $H > H_{c2}$, the first possibility to consider is whether there are any remnants of superconductivity, such as SC fluctuations ..." (please see our reply to comment 1 of reviewer #1).

Therefore, the Discussion section is devoted entirely to the discussion of possible mechanisms responsible for $R_H = 0$ in the $H > H_{c2}$ regime, including the presence of SC fluctuations. The discussion was expanded also in response to some comments of reviewers 1 and 2. In particular, we have argued that several possible mechanisms, including SC fluctuations, are highly unlikely

as the origin of $R_H = 0$. We have also discussed a few promising models, such as the smectic metal model proposed by Emery et. al (ref. 49), and more general models for strongly correlated matter that feature emergent particle-hole symmetry (refs. 51, 52).

Fig 1 combines the newly measured Hall effect and previously published transport measurements. Are these and prior measurements conducted on the same two samples? Is there a variation in the properties of the samples with the same nominal composition?

Some of the data in Fig. 1 were indeed reported previously (refs. 3 and 4), which established the phase boundaries of the vortex solid, vortex liquid, and the high- H anomalous normal state. For those studies, several single crystal samples of each material were studied, and the results agree very well.

A small, quantitative variation of the $\text{La}_{1.7}\text{Eu}_{0.2}\text{Sr}_{0.1}\text{CuO}_4$ sample “B” low-temperature properties was observed after ~ 3 years of measurements and attributed to a small change in the effective doping (sample “B1”). However, the phase diagram remained qualitatively the same (see Supplementary Fig. 8 in ref. 4). For our Hall measurements, the same two $\text{La}_{1.7}\text{Eu}_{0.2}\text{Sr}_{0.1}\text{CuO}_4$ and $\text{La}_{1.48}\text{Nd}_{0.4}\text{Sr}_{0.12}\text{CuO}_4$ samples are used. We note that most of our data were obtained before the $\text{La}_{1.7}\text{Eu}_{0.2}\text{Sr}_{0.1}\text{CuO}_4$ sample change, i.e. on sample B. After the sample change, we repeated the Hall measurements on sample B1 and obtained the same results. We have added the following text in Methods, section “Samples”.

“The same two samples were also studied previously^{3,4}. After ~ 3 years, the low- T properties of the $\text{La}_{1.7}\text{Eu}_{0.2}\text{Sr}_{0.1}\text{CuO}_4$ sample “B” changed, which was attributed to a small change (increase) in the effective doping, but its phases remained qualitatively the same⁴. We repeated the Hall measurements after the sample had changed, and obtained the same results.”

Are samples’ superconducting properties affected by contact annealing at 700 C and 450 C?

To achieve the best possible electrical contacts, which is crucial for our measurements at the ultralow temperatures, we prepared the gold contacts in two steps, as described in Methods. Our extensive characterization of these crystals after both annealing steps confirm that their properties are consistent with those expected from the literature, including charge order transition temperature, superconducting T_c , and its doping dependence.

We also performed SQUID measurements on one crystal which was annealed only at 700 C but not subsequently annealed at 450 C, and the observed T_c was again consistent with the expected value (please see our previous results in ref. 3). Therefore, we found that the sample quality is not affected by the annealing.

In response to the reviewer’s question, we have added the following sentence in Methods, at the end of subsection “Samples”.

“Meanwhile, we found no change in the superconducting properties of the samples before and after the annealing.”

What are the actual values for the current, at least for the low T traces on Extended Figs. 1,2, and 6?

The current excitation was chosen based on our previous nonlinear transport study (ref. 3), where we have established the threshold for Joule heating. Therefore, at the lowest T , we have used the highest possible current excitation without heating up the samples. The detailed values of the current have been added to Methods, subsection “Measurements.”, as follows.

“...was used: 10 μA for 0.019 K (Supplementary Fig. 1d); 100 μA for all measurements in fields up to 12 T (Supplementary Fig. 1a), and for the 0.3 K data in Supplementary Figs. 2 and 3; 316 μA for all other measurements.”

Finally, we thank the reviewer for his/her constructive comments and for recommending the publication of our manuscript in Nature Communications. We have addressed the reviewer’s comments in the revised manuscript, and thus we hope the reviewer finds it suitable for publication.

Other changes:

1) For completeness, we have added ref. 25 and modified text on. p. 6, paragraph 1, so that it reads:

“...in the absence of spin order, or in $\text{La}_{2-x}\text{Sr}_x\text{CuO}_4$ (ref. ²⁵), where the charge order is at best very weak.”

2) To reduce the length of the abstract, we have moved two sentences from the abstract to the beginning of the main text.

REVIEWERS' COMMENTS

Reviewer #1 (Remarks to the Author):

The authors have substantively revised the manuscript and carefully considered all of the reviewer comments in doing so. For non-expert readers, many additions will increase the accessibility and likely impact of the results, particularly the discussion of theoretical frameworks and additional sample characterization details. (Upon re-reading, these added temperature scales on p. 5 could be more clearly presented in tabular form, space permitting.) I support proceeding towards publication of the manuscript, and have only one other minor/optional suggestion:

6) The second paragraph on page 10 (as well as a few other moments in the text) is more than sufficiently hedged (“suggests ... might be ... reasonable ... to speculate whether ... might”); I think the authors are justified in simplifying the language to make a clear claim here.

Reviewer #2 (Remarks to the Author):

The authors have made satisfactory changes to the text and addressed the previous comments I had. I am in favour of publishing this article.

Reviewer #3 (Remarks to the Author):

The authors provided a detailed response to the questions and concerns that were raised in the first review. I find that the authors' comments and edits adequately address the shortcomings of the original version. The clarity and readability of the text and figures have also improved. In my opinion, this manuscript is suitable for publication in Nature Communication.

Reply to the Reviewers' Second Report

We are grateful to the reviewers for reading our manuscript and for their constructive comments that have improved the quality of our paper. Our response to the reviewers' comments is given below (comments in blue, our response in black).

Reviewer #1 (Remarks to the Author):

The authors have substantively revised the manuscript and carefully considered all of the reviewer comments in doing so. For non-expert readers, many additions will increase the accessibility and likely impact of the results, particularly the discussion of theoretical frameworks and additional sample characterization details.

We thank the reviewer for the many helpful suggestions that have helped us to improve our manuscript, and for recommending our paper for publication in Nature Communications.

(Upon re-reading, these added temperature scales on p. 5 could be more clearly presented in tabular form, space permitting.)

We have followed the reviewer's suggestion and presented characteristic temperature scales on p. 5 in tabular form.

I support proceeding towards publication of the manuscript, and have only one other minor/optional suggestion:

6) The second paragraph on page 10 (as well as a few other moments in the text) is more than sufficiently hedged ("suggests ... might be ... reasonable ... to speculate whether ... might"); I think the authors are justified in simplifying the language to make a clear claim here.

That part of the sentence has been modified and shortened in the following way:

"...suggests the possibility that the same mechanism might..."

Reviewer #2 (Remarks to the Author):

The authors have made satisfactory changes to the text and addressed the previous comments I had. I am in favour of publishing this article.

We are grateful to the reviewer for his/her helpful comments and suggestions, and for the positive recommendation.

Reviewer #3 (Remarks to the Author):

The authors provided a detailed response to the questions and concerns that were raised in the first review. I find that the authors' comments and edits adequately address the shortcomings of the original version. The clarity and readability of the text and figures have also improved. In my opinion, this manuscript is suitable for publication in Nature Communication.

We thank the reviewer for his/her helpful comments and suggestions, and for recommending the publication of our manuscript in Nature Communications.